# Identification and Functional Validation of the *PeDHN* Gene Family in Moso Bamboo

**DOI:** 10.3390/plants14101520

**Published:** 2025-05-19

**Authors:** Yaqin Ye, Yanting Chang, Wenbo Zhang, Tiankui Chu, Hanchen Tian, Yayun Deng, Zehui Jiang, Yanjun Ma, Tao Hu

**Affiliations:** 1International Center for Bamboo and Rattan, Beijing 100102, China; yyq68907049@163.com (Y.Y.); changyanting@icbr.ac.cn (Y.C.); wenbozhang@icbr.ac.cn (W.Z.); chutiankui@163.com (T.C.); txmy77106@163.com (H.T.); yayundeng@icbr.ac.cn (Y.D.); jiangzh@icbr.ac.cn (Z.J.); 2Key Laboratory of National Forestry and Grassland Administration, on Bamboo & Rattan Science and Technology, Beijing 100102, China; 3Pingxiang Bamboo Forest Ecosystem Research Station, Pingxiang 532600, China

**Keywords:** DHN, moso bamboo, functional verification, NaCl, ABA

## Abstract

As climate change intensifies soil drought and salinization, enhancing the drought and salt tolerance of moso bamboo (*Phyllostachys edulis*) is urgent. DHN genes are crucial for plant stress responses and have gained attention in plant resistance to drought and salinity. This study identified nine DHN family members (*PeDHN1*–*PeDHN9*) from moso bamboo, which were classified into K_2_S-type, YK_2_S-type, and Y_2_K_2_S-type dehydrins based on their characteristic motifs. We employed integrated bioinformatics approaches to analyze their gene structure, phylogeny, biological properties, and expression patterns under various stress conditions. Five genes *(PeDHN2/4/5/6/8*), which may have significant functional roles in moso bamboo, were selected for cloning. Subcellular localization experiments showed that YK_2_S-type dehydrins (PeDHN2/5/6) localized to both the nucleus and the plasma membrane, while K_2_S-type dehydrins (*PeDHN4*/8) were exclusively localized to the plasma membrane, indicating functional differentiation. qRT-PCR analysis revealed that the expression of *PeDHN2/4/5/6/8* was significantly responsive to stress treatments with ABA, NaCl, and PEG. Additionally, overexpressing these genes in rice significantly enhanced seed germination rates and root development under salt and ABA stress, further confirming that PeDHN2/4/5/6/8 contribute to enhancing plant stress tolerance. Yeast one-hybrid assays demonstrated that two *PeABF1* proteins could bind to the promoter of *PeDHN4*, suggesting that *PeDHN4* may regulate stress responses through the ABA signaling pathway. Thus, these findings demonstrate that PeDHN2/4/5/6/8 are closely related to the response of moso bamboo to drought and saline-alkali environments. This research offers insights for moso bamboo cultivation and theoretical foundations for bamboo genetic improvement in stress environments.

## 1. Introduction

Moso bamboo (*Phyllostachys edulis*), a representative bamboo species, is one of the most multifunctional and renewable resources in the plant kingdom, often referred to as the “green gold of the forest” [1]. In recent years, mounting environmental pressures have driven extensive research into stress-resistant genes and tolerance mechanisms in moso bamboo. Although multiple candidate genes have been identified, the absence of an effective genetic transformation system has hindered functional validation, impeding research and breeding advancements [2,3]. This limitation has led researchers to employ rice—a closely related species with an established transformation system—as a surrogate model for functional validation studies [4,5,6,7].

Among the molecular mechanisms plants employ to cope with environmental stresses such as drought, salinity, and extreme temperatures, dehydrins (DHNs) have garnered widespread attention as important stress-response proteins. Studies have shown a positive correlation between the accumulation of DHNs and the plant’s tolerance to drought, salt, or freezing stress in various species [8,9]. DHNs belong to the late embryogenesis abundant (LEA) protein family and are characterized by conserved sequence motifs: K, Y, and S segments. By definition, dehydrins must contain at least one lysine-rich K segment [10]. Over 20 years ago, Close elucidated the conserved structural architecture of dehydrins by identifying three signature motifs enriched in tyrosine (Y), serine (S), and lysine (K) residues. This discovery established the YSK nomenclature system. Based on the types and numbers of signature motifs containing these characteristic amino acids, DHNs can be divided into five types: Kn, SKn, KnS, YnKn, and YnSKn [11]. The K segment (EKKGIMDKIKEKLPG), a core functional domain of DHNs, is rich in lysine and forms an amphipathic helix that interacts with membranes or proteins, protecting them from stress-induced damage, thereby mediating cellular stress tolerance and maintaining enzyme activity [12]. The Y segment (T/VDEYGNP), with 1–3 copies at the N-terminus, and the S segment (SSSS-SSSSSSSS) can both play roles in transporting DHNs into the nucleus upon phosphorylation [13]. It was later discovered that in many conifers and angiosperms, an F segment (DRGLFDFLGKK) exists between the S and K segments [14].

DHNs have now been identified in a wide range of plants, where they play crucial roles in responding to abiotic stresses like drought, low temperatures, salinity, and oxidative stress. Notable examples include rice (*Oryza sativa*) [15], maize (*Zea mays*) [16], bread wheat (*Triticum aestivum*) [17], and chili pepper (*Capsicum annuum*) [18]. In citrus, the *CuCOR19* dehydrin scavenges hydroxyl and peroxyl free radicals, reducing oxidative damage caused by water stress [19]. In *Populus trichocarpa*, overexpression of the *PtrDHN-3* gene in *Arabidopsis thaliana* significantly increases the activity of superoxide dismutase (SOD) and peroxidase (POD), reducing the accumulation of reactive oxygen species (O^2−^ and H_2_O_2_) and enhancing salt stress tolerance [20]. In *Picea wilsonii*, *PicW1* reduces the aggregation and denaturation of lactate dehydrogenase (LDH) under freezing and high-temperature stress, improving the plant’s stress resistance [21]. However, there has been limited systematic research on the DHN gene family in bamboo species.

In this study, we focused on moso bamboo, identifying the PeDHN gene family and conducting bioinformatics analysis. Based on the number of stress-related cis-regulatory elements in their promoters and expression data, we selected five genes (*PeDHN2/4/5/6/8*) with a higher number of stress-related cis-regulatory elements and higher expression under multiple stress conditions for further functional validation. Expression patterns of these five *PeDHNs* were analyzed in moso bamboo seedlings under ABA, NaCl, and PEG stress treatments to assess their roles in stress responses. Their contributions to stress tolerance were evaluated by overexpressing them in rice. Furthermore, yeast one-hybrid assays confirmed the interaction between *PeDHN4* and two ABF1 proteins. This study not only broadens the genetic resources for stress resistance in bamboo but also provides a theoretical basis for the targeted breeding of salt-alkali-resistant moso bamboo.

## 2. Results

### 2.1. PeDHN Gene Identification and Basic Characterization

By using the amino acid sequences of eight *Arabidopsis* and eight rice DHNs, we performed BLAST 2.9.0 and Hmmer 3.0 searches on the moso bamboo genome. After confirming the presence of functional domains, we identified nine *PeDHN* members in moso bamboo, which were named *PeDHN1*–*PeDHN9* based on their positions on the chromosomes. The *DHN* genes in moso bamboo are distributed unevenly across six scaffolds. Scaffold13 contains two genes, *PeDHN5* and *PeDHN6*, while scaffold2 contains three genes: *PeDHN1*, *PeDHN2*, and *PeDHN3*. The remaining four scaffolds each contain one gene: *PeDHN4* is located on scaffold3, *PeDHN7* on scaffold14, *PeDHN8* on scaffold17, and *PeDHN9* on scaffold22 (Figure 1A).

Based on the identification of the basic physicochemical properties of PeDHN proteins, their transcript lengths range from 387 bp to 1479 bp, with protein lengths spanning from 128 aa to 492 aa. The isoelectric points of these proteins range from 5.23 to 9.85, and their molecular weights vary from 13,223.55 Da to 51,045.33 Da. The Grand Average of Hydropathicity (GRAVY) for the nine PeDHN proteins ranges from −0.021 to −1.241 (Appendix A). In terms of secondary structure, Random Coil (Cc) predominates, with a small fraction containing Alpha helix (Hh) structures (Appendix A). This disordered structure facilitates multifunctionality and dynamic responsiveness, aligning with its role as a dehydrin, while the locally ordered regions may contribute to specific interactions.

Phosphorylation is a critical post-translational modification that alters protein activity, stability, subcellular localization, and interactions, thereby influencing cellular functions. In our analysis of the phosphorylation sites of DHN family proteins, we found that phosphorylation sites were predominantly located in Serine (S), followed by Threonine (Th) and, lastly, Tyrosine (Ty). In particular, PeDHN3 had the highest number of phosphorylation sites, with 48 sites in Serine, 24 in Threonine, and 9 in Tyrosine. Among the kinase binding sites, PKC, CKII, CKI, and CdC2 were the most prevalent in the PeDHN family. All PeDHNs contained one RSK site. PeDHN3, PeDHN4, PeDHN7, PeDHN8, and PeDHN9 had PKA sites, while PeDHN3 and PeDHN7 also contained EGFR sites. PeDHN1, PeDHN2, PeDHN3, PeDHN5, and PeDHN7 had INSR sites, PeDHN3, PeDHN4, and PeDHN8 contained PKG sites, and PeDHN2, PeDHN3, PeDHN4, PeDHN5, and PeDHN8 contained DNAPK sites. Only PeDHN8 and PeDHN9 had p38MAPK and CDK5 sites. PeDHN8 also contained a GSK3 site, and PeDHN5 had an ATM site (Appendix A). These findings indicate that PeDHN family members integrate multiple kinase signals, forming a multi-layered stress response network. For example, PeDHN8, regulated by PKG, p38MAPK, and CDK5, enhances plant adaptability to complex stress conditions. Various kinases, through phosphorylation or regulation of PeDHNs expression, contribute to ROS scavenging (PKG), DNA repair (DNAPK/ATM), metabolic support (INSR), and signal transduction (p38MAPK).

To explore the structural diversity and characteristics of the DHN genes, we constructed gene structure maps, phylogenetic trees, and motif predictions and analyzed the promoter cis-regulatory elements and structural regions of the nine *PeDHNs*. All *PeDHNs*, except *PeDHN3*, consist of one intron and two exons (Figure 1B). Domain analysis revealed that all PeDHNs contain a Dehydrin or Dehydrin superfamily domain, and PeDHN3 also contains a Rhomboid domain (Figure 1C). Motif analysis of the *PeDHN* protein sequences using the MEME tool identified six motifs. Motifs 1 and 2 contain the characteristic K segment, motif 4 contains the Y segment, and motif 3 contains the S segment (Figure 1B,D). Additionally, several motifs that do not conform to the traditional YSKF type were identified in the moso bamboo dehydrin family, suggesting potential additional roles for these motifs within the family. Statistical analysis of the protein motifs revealed that the S segment in moso bamboo consists of 7–9 serine residues, the Y segment is characterized by VDEYGNP, and the K segment by EKKGV\IM\KEKIKEKLPG. All *PeDHNs* contain two K segments and one S segment. *PeDHN4*, PeDHN8, and PeDHN9 are K_2_S-type, while PeDHN1, PeDHN2, PeDHN3, PeDHN5, PeDHN6, and PeDHN7 are YK_2_S-type dehydrins (Table 1). Moreover, the YK2S-type dehydrins contain six motifs, while in the K_2_S-type dehydrins, PeDHN9 lacks motif 5, and *PeDHN4* and PeDHN8 lack both motif 5 and motif 6.

To further investigate the potential biological functions of *PeDHNs*, we analyzed the cis-regulatory elements in the promoter regions of these nine genes. Among the stress-related cis-regulatory elements, the most abundant in *PeDHNs* were ABRE, along with elements responsive to MeJA, ABA, and Salicylic acid hormones (Figure 1C,E). This suggests that the regulation of *DHN* genes in response to stress is likely associated with the metabolic pathways of ABA, MeJA, and Salicylic acid.

### 2.2. Phylogenetic Tree Construction and DHN Gene Synteny Analysis

To obtain the genomic characteristics and evolutionary information of the moso bamboo DHN gene family, we constructed a phylogenetic tree using homologous and genome-annotated protein sequences from the dicot model plant *Arabidopsis thaliana* and the monocot model plant *Oryza sativa*. The goal was to explore the relationship between *PeDHNs* in moso bamboo and those in other species, thereby predicting their potential functions. Phylogenetic analysis revealed that *PeDHNs* can be divided into four specific subgroups. PeDHN9 formed an independent cluster. PeDHN7 clustered with *AtLEA*, while PeDHN5 exhibited close phylogenetic affinity with *OsLEA25*. PeDHN4 and PeDHN8 were grouped together and may share functional similarity with *OsLIP9*. Additionally, PeDHN1, PeDHN2, PeDHN3, PeDHN5, and PeDHN6 constituted another cluster. Notably, PeDHN3 appears to be functionally analogous to *OsLEA25*, whereas PeDHN6 may exhibit functional parallels with *OsRab16A*. Collectively, most DHN genes in moso bamboo demonstrated closer evolutionary relationships with *Oryza sativa* within the Poaceae family (Figure 2A).

Collinearity within the moso bamboo genome, as well as between other bamboo genomes and the rice genome, was analyzed using the MCScanX software in TBtools v2.225. The results showed that there are two pairs of collinear genes (*PeDHN8* and *PeDHN4*, *PeDHN5* and *PeDHN6*) of *PeDHNs* in moso bamboo (Figure 2B), which may be related to the two rounds of genome duplication events in moso bamboo [22]. Interspecific collinearity analysis was also performed on *Bonia amplexicaulis*, *Olyra latifolia*, and *Oryza sativa*. There are three pairs of collinear genes between moso bamboo and *Olyra latifolia*, five pairs between moso bamboo and *Bonia amplexicaulis*, and eleven pairs between moso bamboo and rice (Figure 2C).

### 2.3. Transcriptome Data Analysis of PeDHNs Under Various Treatment Conditions

To further investigate the expression profiles of DHN genes under various stress conditions, we performed transcriptome data analysis using previously published moso bamboo transcriptome data and data from the National Center for Biotechnology Information (NCBI) database (PRJNA715101) [23]. The results revealed that *PeDHN1*, *PeDHN2*, *PeDHN4*, *PeDHN5*, *PeDHN6*, and *PeDHN8* displayed elevated expression levels under PEG, NaCl, ABA, and dehydration treatments. Notably, PeDHN3 exhibited a 20-fold increase in expression only under dehydration treatment (2 h), with no significant changes under other treatments. *PeDHN4* and *PeDHN8* were responsive to SA and cold treatments. *PeDHN7* demonstrated a significant increase in expression after 24 h of PEG treatment, 24 h of NaCl treatment, and under multiple treatments at 0 h and dehydration conditions. *PeDHN9* showed minimal expression changes under PEG, NaCl, SA, ABA, and the combined treatments of PEG, NaCl, SA, and ABA, but expression was significantly increased under dehydration treatment. Additionally, *PeDHN3* showed almost no expression under PEG, NaCl, SA, and ABA treatments (Figure 3).

### 2.4. Subcellular Localization of DHN Gene Family Members

Subcellular localization predictions from the WOLF PSORT server indicated that, except for PeDHN3, which is localized to the plasma membrane, endoplasmic reticulum, nucleus, and golgi apparatus, and PeDHN9, which is localized to the nucleus and chloroplast, the remaining PeDHNs are all localized in the nucleus (Appendix A). Based on the number of stress-related cis-regulatory elements in their promoters and transcriptome analysis results, five genes with higher expression levels under various stress conditions were selected for cloning and the construction of pHG fusion expression vectors: PeDHN2, PeDHN4, PeDHN5, PeDHN6, and PeDHN8. The cloning and construction of the pHG fusion expression vectors were completed. Using pHG fusion expression vectors tagged with GFP, the subcellular localization of the five PeDHNs proteins was investigated. The results showed that the three YK_2_S-type dehydrins, PeDHN2, PeDHN5, and PeDHN6, were localized to both the nucleus and the plasma membrane, while the two K_2_S-type dehydrins, PeDHN4 and PeDHN8, were exclusively localized to the plasma membrane, with no expression observed in the nucleus (Figure 4). These findings suggest that there are differences in the subcellular localization of YK_2_S-type and K_2_S-type dehydrins, which may be closely related to their molecular structures and functional specialization.

### 2.5. Expression Pattern Analysis of Five PeDHN Genes Under Various Stress Treatments

To investigate the expression patterns of *PeDHNs* under ABA, NaCl, and PEG treatments, we selected *PeDHN2*, *PeDHN4*, *PeDHN5*, *PeDHN6*, and *PeDHN8*, which exhibited higher expression levels under multiple stress conditions, for qRT-PCR analysis based on transcriptome data.

Under 100 µM ABA treatment, the expression patterns of the five *PeDHN* genes varied across different time points (0 h, 1 h, 3 h, 6 h, 9 h, 12 h, and 24 h). PeDHN2, PeDHN5, and PeDHN6 exhibited oscillatory response patterns. These genes showed a rapid increase in expression after 1 h of stress treatment, with expression significantly upregulated compared to the control group. However, their expression levels declined at 3 h (Figure 5A,C). Under prolonged stress, between 6 h and 9 h, these genes showed a secondary activation, with expression gradually rising to a peak, followed by a decrease at 12 h. After 24 h, the expression levels rebounded and stabilized at a higher expression level. *PeDHN4* and *PeDHN8* displayed similar trends, with rapid responses at 1 h, stable expression between 3 h and 12 h, and higher expression at 24 h (Figure 5B,E). Notably, *PeDHN5* and *PeDHN6* exhibited the highest expression levels among the five *PeDHNs* under ABA treatment (Figure 5C,D).

Under 300 mM NaCl treatment, after 0 h, 1 h, 3 h, 6 h, 9 h, 12 h, and 24 h, the five *PeDHNs* displayed similar expression trends. Between 1 h and 12 h, expression levels of all five *PeDHNs* gradually increased, reaching a peak at 12 h. However, at 24 h, expression levels began to decline. *PeDHN5* and *PeDHN6* exhibited the highest expression levels in response to NaCl treatment among the five *PeDHNs* (Figure 5C,D).

Under 20% PEG treatment, after 0 h, 1 h, 3 h, 6 h, 9 h, 12 h, and 24 h, all five *PeDHNs* exhibited a similar pattern of initially increasing, then decreasing, and later increasing again. *PeDHN2* showed an increase at 1 h, a decline at 3 h and 6 h, and a gradual increase from 9 h to 24 h (Figure 5A). *PeDHN4* and *PeDHN8* displayed similar expression trends (Figure 5B,E). From 1 h to 9 h, the expression of *PeDHN4* and *PeDHN8* gradually increased, but at 12 h, a sudden decrease occurred, followed by an increase at 24 h. *PeDHN5* and *PeDHN6* exhibited a similar pattern, with a brief increase at 1 h, a decrease at 3 h, and a gradual increase thereafter (Figure 5C,D). Notably, *PeDHN2*, *PeDHN5*, and *PeDHN6* showed the highest expression levels under 20% PEG treatment compared to the other *PeDHNs*.

### 2.6. Evaluation of Stress Response Characteristics in DHN Transgenic Rice

To investigate the biological functions of *PeDHN2/4/5/6/8* genes, we constructed overexpression vectors containing the target genes and transformed them into wild-type japonica rice via Agrobacterium-mediated transformation. After hygromycin selection and PCR validation with marker primers, 52 homozygous T1 transgenic lines were successfully generated (Appendix A). To assess seed germination and early seedling growth of the transgenic lines under various stress conditions, we cultured the transgenic rice in dual-germination bags using hydroponics. Ten rice seeds were placed in each germination bag, and NaCl (80 mM) and ABA (8 µM) were used as stress treatments. The experimental groups were labeled as OE-*PeDHN2/4/5/6/8* and wild-type (WT) (Appendix A).

The germination rates were observed and recorded on days 7 and 14. Under normal conditions, the germination rates of both WT and transgenic rice seeds were close to 100%, indicating good germination ability. However, under salt stress, the germination rate of WT decreased significantly to 25%, indicating sensitivity to salt stress. In contrast, the germination rates of *OE-PeDHN5* and *OE-PeDHN6* reached 70% and 95%, respectively, which were significantly higher than WT. Under 8 µM ABA stress, the germination rate of WT decreased to 55% on day 7, showing sensitivity to ABA stress, while the germination rates of *PeDHN2/4/6/8* remained between 80% and 95%, much higher than the control group (Figure 6A). Notably, PeDHN6 exhibited a high germination rate in all stress treatments, almost unaffected by salt and ABA stress, with germination rates similar to those under normal conditions. This result suggests that *PeDHN6* has a significant advantage in enhancing rice stress tolerance and may play a key role in regulating seed germination and stress resistance in rice. To further explore the effects of NaCl and ABA on the overexpressing *PeDHN* transgenic rice, we measured the root lengths of rice at days 7 and 14 after treatment. Under normal conditions, *OE-PeDHN2* exhibited significantly longer roots on day 7. By day 14, *OE-PeDHN4* also showed a root length advantage. Under NaCl stress, root growth was inhibited in all treatment groups, though there were no statistical differences between the groups. In the ABA treatment group, significant differences were observed between the transgenic and WT lines on days 7 and 14. On day 14, the overall root lengths were ranked from largest to smallest as follows: *OE-PeDHN2* > *OE-PeDHN4* > *OE-PeDHN8* > *OE-PeDHN5* > *OE-PeDHN6* > WT (Figure 6B).

These results suggest that overexpression of *PeDHNs* may alleviate the inhibitory effects of ABA on root growth by reducing the sensitivity of rice roots to ABA.

After 50 days of continuous NaCl treatment, the WT plants showed wilting and death in their aerial parts, while several overexpression lines exhibited longer roots and vigorous leaf growth. Similarly, after 50 days of continuous ABA treatment, no significant phenotypic differences were observed in the aerial parts of the control and transgenic lines. However, the transgenic lines had significantly longer and thicker roots than the controls, indicating that the overexpression of *PeDHN2/4/5/6/8* enhanced rice resistance to NaCl and ABA (Figure 6C,D).

### 2.7. PeDHN4 as a Target of PeABF1 in the ABA Signaling Pathway

In this study, to investigate the expression regulation mechanism of moso bamboo *DHNs* in response to abiotic stress, we constructed a moso bamboo cDNA yeast library (Appendix A) and cloned the ABRE, ABRE4, DRE core, MYB/b, and MYB Recognition site cis-regulatory elements from the *PeDHN4* promoter sequence, which are associated with stress resistance, into the pHIS2 vector (Figure 7A). We used the yeast one-hybrid system to screen for upstream transcription factors that interact with the *PeDHN4* gene promoter. Sequencing of positive clones from the screen identified 1690 potential genes that interact with the *PeDHN4* tandem promoter sequence. Gene Ontology (GO) enrichment analysis revealed that these genes were primarily involved in processes related to response to stimuli and cellular processes (biological process), associated with the nucleus, intracellular anatomical structures, and cellular components (cellular component), and related to kinase activity, transferase activity, transfer of phosphorus-containing groups, and DNA-binding transcription factor activity (molecular function) (Appendix A). Kyoto Encyclopedia of Genes and Genomes (KEGG) enrichment analysis showed that these genes were enriched in pathways related to stress resistance, such as environmental adaptation, plant–pathogen interactions, oxidative phosphorylation, and environmental information processing (Appendix A). These results indicate that genes interacting with the *PeDHN4* tandem promoter sequence are involved in stress resistance at multiple biological levels. These genes may regulate gene expression and intracellular signaling and contribute to the plant’s response to and adaptation to environmental stress. These findings provide valuable insights into the role of the *PeDHN4* promoter in plant stress-resistance mechanisms.

We subsequently screened six transcription factors related to stress resistance from the 1690 genes identified as interacting with the *PeDHN4* promoter, including three *ABF1*, two *ABF2*, and one *DREB3*. We cloned two *ABF1* genes (*PH02Gene27949* and *PH02Gene02526*), named *PeABF1-1* and *PeABF1-2*, and constructed them into the pGADT7 vector for yeast one-hybrid assays to confirm their interactions with the *PeDHN4* promoter. The results showed that *PeABF1-1* and *PeABF1-2* both interacted with the *PeDHN4* promoter. The growth of pHIS2-*PeDHN4* + pGADT7-*PeABF1-1* was stronger than that of pHIS2-*PeDHN4* + pGADT7-*PeABF1-2*, indicating that *PeABF1-1* binds more efficiently to the *PeDHN4* promoter than *PeABF1-2* (Figure 7B).

## 3. Discussion

This study identifies the PeDHN gene family in moso bamboo and investigates five *PeDHN* genes with potential roles in stress resistance.

Dehydrins are intrinsically disordered proteins lacking well-defined secondary and tertiary structures. These proteins are highly hydrophilic and contain many charged amino acids, enabling them to bind to membranes as peripheral membrane proteins. While they do not exhibit a defined structure in solution, they acquire structure upon binding to ligands (e.g., membranes), and their oligomeric state may change when binding to ions [10]. Among PeDHNs, longer dehydrins (e.g., 492 aa) may contain more repetitive sequences (such as Y/S/K segments), which enhance binding to water, ions, or biomolecules. Shorter isoforms (e.g., 128 aa) may function through specific motifs (such as the conserved K segment) to perform core protective functions. Additionally, the negative GRAVY values of all PeDHNs indicate their high hydrophilicity, a characteristic typical of dehydrins. These structural features enable dehydrins to perform multiple cellular functions and provide the foundation for post-translational modifications. Increasing evidence suggests that dehydrins undergo post-translational modifications, with phosphorylation being the most prominent [24,25]. For example, *PeDHN8* is regulated by PKG, p38MAPK, and CDK5, which enhance plant adaptation to complex stress conditions through phosphorylation. Furthermore, different kinases participate in various cellular processes by phosphorylating or regulating the expression of *PeDHNs*: PKG is involved in ROS scavenging, DNAPK and ATM in DNA repair, INSR in metabolic support, and p38MAPK in signal transduction.

Phylogenetic and synteny analysis showed that *PeDHNs* are more closely related to other *Poaceae* species, and moso bamboo shares more syntenic gene pairs with rice, likely due to rice retaining more ancestral genomic traits, while species like *Bonia amplexicaulis* and *Olyra latifolia* have undergone more genomic rearrangements. In synteny analysis of the moso bamboo genome, *PeDHN4* and *PeDHN8*, as well as *PeDHN5* and *PeDHN6*, were identified as syntenic gene pairs. These pairs share structural similarities and similar stress-responsive expression patterns. PeDHN4 and PeDHN8 are K_2_S-type dehydrins, while PeDHN5 and PeDHN6 are YK_2_S-type dehydrins. Their expression patterns and levels in both moso bamboo and transgenic rice are similar, suggesting they may have evolved from a common ancestor gene.

In studies of other species, differences in DHN characteristic segments affect gene localization. For instance, ZmDHN13 protein in maize is localized to the nucleus, depending on the phosphorylation of the S segment, possibly having DNA repair or protection functions. In wheat, the cold-induced WCS120 Kn-type DHN is localized to both the cytoplasm and nucleus due to the presence of the Nuclear Localization Signal (NLS) segment [13,26]. In contrast, the NLS and S segments of *MsDHN1* in *Medicago sativa* do not contribute to nuclear localization, suggesting that nuclear entry occurs by eliminating FS or FSK segments [27]. In this study, three YK_2_S-type dehydrins (PeDHN2, PeDHN5, and PeDHN6) were expressed in both the nucleus and plasma membrane, whereas two K_2_S-type dehydrins (PeDHN4 and PeDHN8) were expressed only in the plasma membrane (Figure 4). This may be due to the absence of motif 5 and motif 6, which may prevent their entry into the nucleus. Future experiments could involve truncating DHN sequences and constructing fluorescent fusion vectors to explore the impact of characteristic segments on functional localization.

Arid regions experience scarce rainfall and intense evaporation. As soil moisture evaporates, salts like sodium, calcium, and magnesium sulfates and chlorides accumulate in the surface soil due to lack of leaching by rain, causing soil salinization. Obviously, there is a close relationship between aridity and salinization. Plants can significantly enhance stress resistance through mediation by compounds like ABA, with ABA being a key signaling molecule that mitigates oxidative damage to plant organs and tissues under stress [28,29]. ABA signaling includes both dependent and independent pathways: The dependent pathway involves the ABA receptor PYR/PYL/RCAR and the PP2C gene family [30,31], which activate ABA response genes such as members of the *AREB/ABF* subfamily [32,33]. In the independent pathway, transcription factors such as *NAC*, *DREB*, and *AP2/ERF* play direct roles [34]. In the PeDHN gene family, we found that the ABRE cis-regulatory element is predominant among stress-related elements. As a key component of ABA signaling, ABRE binds transcription factors to activate downstream gene expression [35]. Many *DHNs* have been reported to be ABA-inducible, serving as functional genes in abiotic stress responses [36,37]. In potato, the *AAO* gene promoter region contains multiple ABRE elements, which are core functional domains responding to ABA, and binding to transcription factors directly activates gene expression [38]. In this study, transgenic *OE-PeDHN* rice showed higher germination rates and longer root lengths under NaCl and ABA treatments compared to WT rice, indicating that these genes function in stress responses. After 50 days of stress treatment, the root systems of the NaCl and ABA treatment groups were significantly thicker compared to WT rice (Figure 6C,D), suggesting that these genes may promote stress response by enhancing root growth. Moreover, the expression of the five *PeDHNs* genes in the leaves of moso bamboo significantly increased under ABA and NaCl treatments, which aligns with the transcriptome data trends. It is hypothesized that, when subjected to abiotic stress, *DHN* genes are induced to upregulate and improve the plant’s tolerance to salt stress by promoting root growth.

Currently, there is limited research on the mechanisms by which *DHN* genes respond to stress. In wheat, transcription factors *bHLH47-like*, *bHLH49-like*, and *AN1* interact with the promoter of the *WZY2* dehydrin gene, and WZY2 protein also interacts with the PP2C protein in the ABA signaling pathway to regulate the expression of stress response genes [39]. In *Medicago sativa*, the *MsDHN1* promoter interacts with *MsABF*, and this gene also interacts with two water channel proteins [27]. In this study, we demonstrated the interaction of *PeDHN4* with two *ABF1* transcription factors using the yeast one-hybrid assay. *ABF1* belongs to the A subfamily of the *bZIP* transcription factor family and is a core regulatory component of the ABA signaling pathway. By recognizing and binding to the ABRE, *ABF1* activates the expression of stress-resistant genes, enhancing plant adaptation to adverse environments such as drought and salinity [38,40,41]. The DHN gene family contains many ABRE elements, and the binding of *ABF1* to the *PeDHN4* promoter suggests that signals from abiotic stress may be transmitted to *DHN* genes through *ABF1* in the ABA signaling pathway. *DHNs* may improve plant stress resistance by promoting root growth and other mechanisms. In the future, if we can perform transient transformation of PeDHN4 in rice ABA signaling mutants, we can further reveal the role of PeDHN4 in the ABA metabolic pathway.

Our study comprehensively analyzes the molecular features and expression patterns of the PeDHN gene family. We cloned five *PeDHN* genes associated with stress resistance and found that they accumulate under NaCl, ABA, and PEG treatments. Among these, *PeDHN4* is a target of two *ABF1* genes and participates in the ABA metabolic pathway. This provides a theoretical basis for studying the stress-resistance mechanisms and molecular breeding of moso bamboo.

## 4. Materials and Methods

### 4.1. Plant Materials

In this study, we used moso bamboo seeds (collected from Guilin, Guangxi, China) as the research materials and grew them into seedlings in a controlled environment chamber. The seedlings were grown in a controlled environment with a temperature of 25 °C, relative humidity of 80%, and a 16 h light/8 h dark cycle until they were 6 months old. Sixty seedlings of similar size and growth were selected, and after washing the roots, the seedlings were transferred to Kimura’s nutrient solution for 4 weeks of hydroponic cultivation. Stress treatments were applied using Kimura’s nutrient solution supplemented with 20% PEG-6000 (to simulate drought), 300 mM NaCl (for salt stress), and 100 μM ABA (hormonal treatment). Leaf samples were collected at 0, 1, 3, 6, 9, 12, and 24 h, and were immediately frozen in liquid nitrogen and stored at −80 °C for further analysis.

Similar experiments with transgenic rice were conducted under the growth conditions of 24 ± 1 °C, 75% relative humidity, light intensity of 40 μmol·m^−2^·s^−1^, and a 16 h light/8 h dark cycle. Positive T1 generation rice seeds were placed in germination bags for stress sensitivity testing using Kimura’s nutrient solution supplemented with either 80 mM NaCl or 8 μM ABA.

### 4.2. Identification of the DHN Gene Family in Moso Bamboo

The genome, coding sequences (CDS), protein sequences, and annotation files for moso bamboo were retrieved from the BambooGDB database (https://bioinformatics.cau.edu.cn/bamboo/index.html, accessed on 24 October 2023). Information on the genome and protein sequences of *Arabidopsis thaliana* was obtained from the TAIR10 database (https://www.arabidopsis.org/, accessed on 24 October 2023), while information for *Oryza sativa* was sourced from the Ensembl Plant database. Basic Local Alignment Search Tool 1.4.0 (BLAST) and Hidden Markov Model Enhanced for Remote homology detection 3.3.2 (HMMER) searches were performed using DHN amino acid sequences from *Arabidopsis thaliana* and *Oryza sativa* as queries. After the identification of sequences common to both searches and the removal of duplicates, Simple Modular Aarchitecture Research Tool (SMART) analysis was conducted using the NCBI Conserved Domain Database v3.21 (https://www.ncbi.nlm.nih.gov/cdd/, 24 October 2023) and Tbtools v2.225 in the Protein Families Database (PFAM) database 37.3 (http://pfam-legacy.xfam.org/, accessed on 24 October 2023) to confirm the presence of dehydrin domains in the sequences [42].

### 4.3. Protein Structures, Biochemical Properties, and Phylogenetic Analysis of PeDHNs

The physicochemical properties of the PeDHN proteins were analyzed using ProtParam (https://web.expasy.org/protparam/, accessed on 24 October 2023) [43]. The secondary structures of the PeDHN were examined using Network Protein Sequence @nalysis (NPS@) 2.16.0 (https://npsa.lyon.inserm.fr/cgi-bin/npsa_automat.pl?page=/NPSA/npsa_sopma.html, accessed on 24 October 2023) [44]. The presence of phosphorylation sites in the proteins was determined using NetPhos 3.1 (https://services.healthtech.dtu.dk/services/NetPhos-3.1/, accessed on 24 October 2023). A phylogenetic tree was constructed using the neighbor-joining method in MEGAX (parameters: bootstrap, 1000 iterations) and was visualized with Interactive Tree Of Life (iTOL) v7 (https://itol.embl.de/, 24 October 2023). Inter-species synteny between moso bamboo, *Bonia amplexicaulis*, *Olyra latifolia*, and *Oryza sativa* was analyzed using MCScanX in TBtools v2.225.

Motifs in the PeDHN proteins were predicted using the Multiple Em for Motif Elicitation 5.5.7 (MEME) online tool and were compared between the PeDHNs and homologous DHN proteins, setting the number of motifs to 10 (https://meme-suite.org/meme/, accessed on 31 October 2024) [45]. The saved XML and Newick files obtained during the construction of the phylogenetic tree were entered into Gene Structure View (Advanced) in TBtools v2.225. Finally, visualization and analysis were performed using TBtools v2.225 (South China Agricultural University, China) [42].

The promoter region 2000 bp upstream of the start codon of the *PeDHN* genes was analyzed for the presence of cis-regulatory elements using PlantCare (https://bioinformatics.psb.ugent.be/webtools/plantcare/html/, accessed on 30 October 2024). Stress-related cis-regulatory elements were selected, and a heatmap was generated.

### 4.4. Transcriptome Expression Profiles of PeDHNs Under Various Treatments

The transcriptomic data of the *PeDHN* genes in moso bamboo leaves following treatment with PEG, NaCl, SA, or ABA, singly and in combination, together with cold and dehydration treatments, were selected based on previously published transcriptomic data. In the data from PRJNA715101, salt and drought stress were simulated using 200 mM NaCl and 25% PEG, respectively, in nutrient solution, with exposure for 3 and 24 h. Nutrient solutions containing 1 mM and 1 μM concentrations of SA and ABA for 3 and 24 h, respectively, were used for SA and ABA treatments. Control moso bamboo seedlings were cultivated in 1/4 Hoagland nutrient solution under conditions of 25 ± 2 °C, 70 ± 10% relative humidity, and a natural light cycle.

The dehydration and cold treatments used for the moso bamboo seedlings were performed as previously described [23].

The transcriptomic data were uniformly converted to TPM, and log2(TPM + 1) values were used to generate heatmaps.

### 4.5. Gene Cloning and Vector Construction

RNA was extracted from moso bamboo leaves using a Quick RNA Isolation Kit (Hua Yueyang, Beijing, China) as directed. The RNA concentration was measured using a microvolume spectrophotometer (Thermo Fisher Scientific, Waltham, MA, USA), and the integrity and quality were assessed using 1% agarose gel electrophoresis (Bio-Rad, Hercules, CA, USA). First-strand cDNA was synthesized using the extracted RNA as a template with a reverse transcription kit (TaKaRa, Kusatsu, Japan). Appropriate primers were designed based on primer design principles using SnapGene 6.0.2 (San Diego, CA, USA) (Appendix A). The target sequences were amplified using 2× Phanta^®^ Max Master Mix (Vazyme, Nanjing, China), and the amplified products were purified using a DNA purification and recovery kit (Tiangen, Beijing, China). The target sequence was inserted into the pCE2TA/Blunt-Zero vector (Vazyme, Nanjing, China) using a 5 min TA/Blunt-Zero Cloning Kit (Vazyme, Nanjing, China) in a metal bath at 25 °C. The ligation product was transformed into *E. coli* DH5α competent cells (Coolaber, Beijing, China) for colony PCR. Positive colonies were selected and sent to Suzhou Azenta Company (Suzhou, China) for Sanger sequencing of the DNA. Sequence analysis was performed using SnapGene 6.0.2. Finally, the plasmid was extracted using a plasmid extraction kit (Tiangen) and stored at −20 °C for future use.

### 4.6. Subcellular Localization of PeDHNs

The constructed expression vector plasmids and the cell membrane localization marker pCAMBIA1300-35S-PM-mCherry were transformed into Agrobacterium competent cells (GV3101, Weidibio, Shanghai, China) using the heat-shock method. The cells were cultured on LB solid medium containing kanamycin (50 μg/mL) and rifampicin (50 μg/mL) at 28 °C for 2 days. Positive colonies were identified by PCR and were subsequently inoculated into 25 mL of LB liquid medium and cultured at 28 °C with shaking at 200 rpm for 16 h. The bacterial cells were then collected by centrifugation at 4000 rpm for 10 min at room temperature, and the supernatant was discarded. The cells were resuspended in suspension buffer (10 mM MES-KOH, 10 mM MgCl_2_, 200 μM acetosyringone, pH 5.7) to achieve an optical density (OD600) of approximately 0.6. After incubation for 3 h, the bacterial suspensions containing the plant expression vector plasmid and the cell membrane localization marker were mixed in a 1:1 ratio. The infection solution was then injected into the dorsal surfaces of tobacco leaves using a 1 mL syringe. The infected plants were watered thoroughly and incubated in the dark at 25 °C for 1 day, followed by 1–2 days of light exposure. The infected tobacco leaves were cut into 1 cm^2^ squares and stained with 4′,6-diamidino-2-phenylindole (DAPI) solution (10 mg/mL) at room temperature for 10–15 min, and then washed five times with sterile water. The fluorescence signals of EGFP, mCherry, and DAPI were observed and photographed using the Axio Imager M2 microscope (Zeiss, Oberkochen, Germany) with excitation wavelengths of 465 nm, 610 nm, and 509 nm, respectively.

### 4.7. Real-Time Quantitative Reverse Transcription PCR of Five PeDHN Genes Following Stress Treatment

Seven-month-old moso bamboo seedlings were treated with Kimura’s nutrient solution alone, and nutrient solutions containing 80 mM ABA, 100 mM NaCl, or 20% PEG for 0, 3, 6, 9, 12, and 24 h. At each time point, the second leaf from the top was collected for real-time quantitative reverse transcription PCR (RT-qPCR) analysis. Quantitative PCR primers for the *PeDHN* genes were designed using Primer Premier v.6 (Appendix A) [46]. RT-qPCR was performed using TB Green & Premix Ex TaqTM II (TaKaRa) on a qTOWER RT-PCR system (Analytik, Jena, Germany). *GAPDH* (*PH02Gene12335*) was used as the reference gene, and the relative gene expression in moso bamboo leaves under three treatments and six time points was analyzed using the 2^−ΔΔCT^ method [47], with expression levels in the 0-day moso bamboo leaf tissue representing the control.

### 4.8. Genetic Transformation of Rice

Seamless cloning primers were designed using SnapGene 6.0.2 (Appendix A), with the pCE2TA/Blunt-Zero vector plasmid containing the *PeDHN* fragments used as the template [48]. The *PeDHN* genes with pHG homologous arms were cloned using the 2× Phanta Max Master Mix high-fidelity enzyme kit (Vazyme, Nanjing, China). The CDS of PeDHN2/4/5/6/8 were then inserted into the pHG vector containing an enhanced green fluorescent protein (EGFP) tag using seamless cloning, resulting in the plant expression vectors 35S::*PeDHN2*–EGFP, 35S::*PeDHN4*–EGFP, 35S::*PeDHN5*–EGFP, 35S::*PeDHN6*-EGFP, and 35S::*PeDHN8*–EGFP. The constructed plant expression vector plasmids were heat-shock transformed into Agrobacterium EHA105 competent cells (Boyuan, Wuhan, China) and sent to Boyuan BioTech Co. (Wuhan, China) for genetic transformation of japonica rice (cv. Nipponbare). Mature rice grains that were free from mold contamination and had intact embryos were chosen. The grains were disinfected in 75% alcohol for 1 min, then rinsed with sterile water. Next, they were disinfected in 15% sodium hypochlorite for 20 min, followed by three rinses with sterile water, each lasting 1 min. The sterilized grains were inoculated onto the callus induction medium and incubated at 26 °C under a 16 h photoperiod for 20 days. The developed calli were infected with Agrobacterium competent cells suspension carrying the target plasmid, then transferred to the screening medium and incubated at 26 °C in the dark for 20 to 30 days. The positive calli were moved to the secondary screening medium and cultured under the same conditions for an additional 7 to 10 days. The calli were transferred to the differentiation medium and cultured at 25 to 27 °C with light for 15 to 20 days. Once shoots reached 2 to 5 cm in length, they were transferred to the rooting medium and cultured at 30 °C with light for 7 to 10 days. Finally, gel electrophoresis analysis with hygromycin-specific detection primers confirmed 15 transgenic japonica rice seedlings, all exhibiting positive PCR amplification bands. Seeds collected from these plants after full cultivation cycles were subsequently used for functional validation studies.

### 4.9. Yeast Library Screening and Yeast One-Hybrid Assays

Moso bamboo seedlings that had undergone drought stress, together with normal moso bamboo seedlings, flowers, and immature embryos, were collected and stored in liquid nitrogen at −80 °C. The samples were sent on dry ice to Ruiyuan BioTech Co. (Nanjing, China) for the construction of a moso bamboo cDNA yeast library (Appendix A). The cis-regulatory elements in the *PeDHN4* promoter were synthesized into a 210 bp sequence and inserted into the pHIS2 vector. The pHIS2 vector containing the *PeDHN4* promoter was transformed into the Y187 yeast strain, followed by the introduction of the moso bamboo cDNA library plasmids. The plasmids were then plated on SD-TLH + 75 mM 3AT plates. Positive clones were selected, and cells were collected by centrifugation, followed by colony PCR. The PCR products were sequenced using next-generation sequencing. The sequencing results were aligned with the annotated moso bamboo genome, and GO and KEGG enrichment analyses were performed to identify transcription factors related to stress tolerance.

Specific primers for the *PeABF1* (*PH02Gene02526*) and *PeABF1* (*PH02Gene27949*) genes were designed using Primer Premier v.6 software and cloned into the pGADT7 expression vector (Appendix A). The successfully sequenced pHIS2-*PeDHN4* plasmid was co-transformed into *Saccharomyces cerevisiae* Y187 with pGADT7-*ABF1-1* (*PH02Gene27949*) and pGADT7-*ABF1-2* (*PH02Gene02526*). Positive control plasmids (pHIS2-P53 + pGADT7-m53) and negative control plasmids (pHIS2-P53 + pGADT7-empty) were used for comparison. The co-transformed yeast cells were plated on SD/-Leu/-Trp (DO) plates containing different concentrations of 3AT and were incubated at 30 °C for 3 days. No colony growth was observed on the 75 mM 3AT plates, indicating no activation of the HIS3 reporter gene. The yeast was then transferred to SD/-Leu/-Trp/-His (1/2DO) medium containing 75 mM 3-AT and incubated at 30 °C for 3 days, followed by recording and imaging of the results.

## 5. Conclusions

In this study, we identified nine members of the DHN gene family in moso bamboo and cloned five of them. *PeDHN2*, *PeDHN5*, and *PeDHN6* were localized to both the nucleus and plasma membrane, while *PeDHN4* and PeDHN8 were localized exclusively to the plasma membrane. The expression of *PeDHN2*, *PeDHN4*, *PeDHN5*, *PeDHN6*, and *PeDHN8* was induced by NaCl and ABA, and all these genes enhanced rice resistance to NaCl and ABA. Two transcription factors, *ABF1-1* and *ABF1-2*, which bind to the *PeDHN4* promoter, were identified, and preliminary validation suggests that *PeDHN4* may regulate the response to drought and salt-alkali stress through the ABA signaling pathway. These findings provide valuable insights into the roles of PeDHNs in enhancing plant tolerance to abiotic stress and offer potential mechanisms for their function.

## Figures and Tables

**Figure 1 plants-14-01520-f001:**
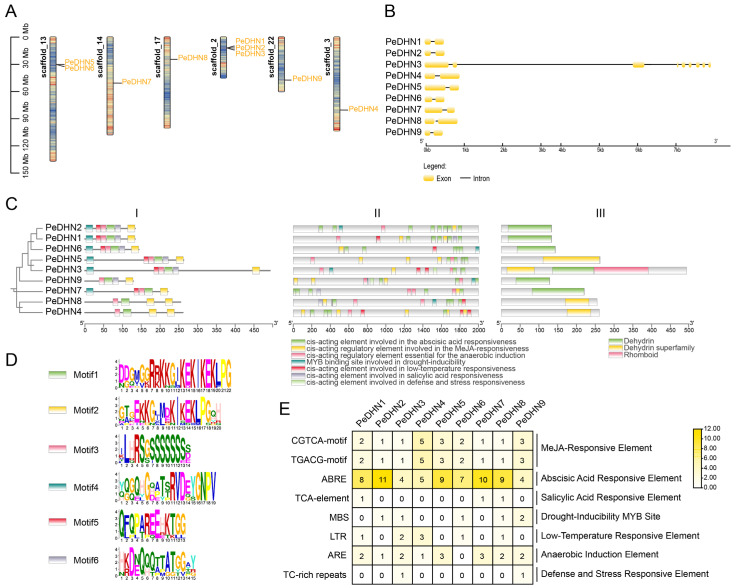
Structural features of the *PeDHN* genes. (**A**) The chromosomal locations of *PeDHNs*. Gene density is shown as a heatmap on the scaffolds, with high gene density represented in red and low density in blue. (**B**) Gene structure of *PeDHNs*. Exons are represented in yellow, and introns are shown as straight lines. (**C**) Phylogenetic tree of PeDHNs and conserved amino acid sequence analysis (**I**), distribution of stress-responsive cis-regulatory elements involved in abiotic stress tolerance in the 2000 bp upstream of the ATG initiation codon of the PeDHNs promoter (**II**), and conserved domains in the coding region of PeDHNs (**III**). The conserved amino acid sequence types are annotated in panel (**D**). (**D**) Logo diagram of conserved amino acid sequences. (**E**) Heatmap showing the number of stress-related cis-regulatory elements in *PeDHNs*.

**Figure 2 plants-14-01520-f002:**
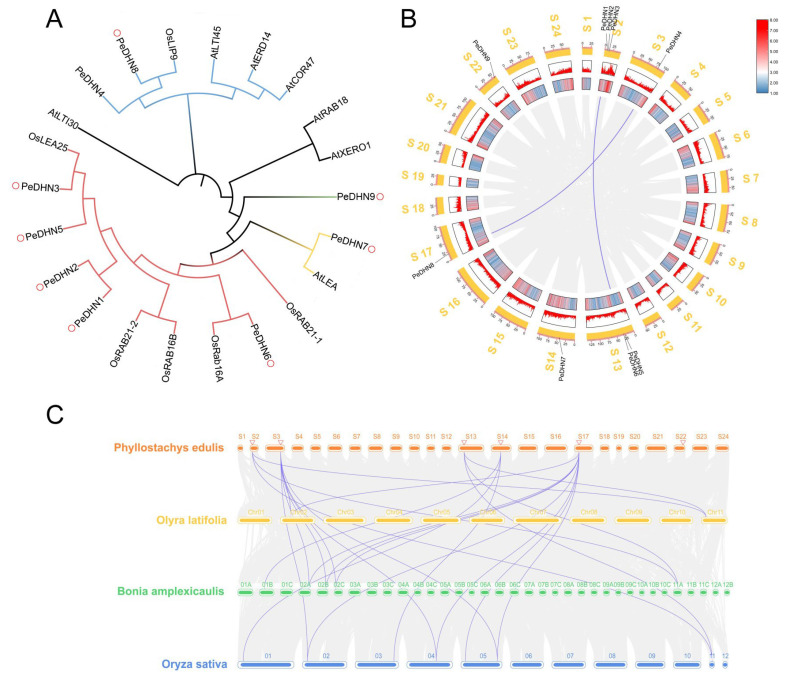
Phylogenetic and synteny analysis of moso bamboo and related species. (**A**) Phylogenetic analysis of 11 species based on moso bamboo using the neighbor-joining method. (**B**) Synteny analysis within moso bamboo species. (**C**) Synteny analysis between moso bamboo and *Bonia amplexicaulis*, *Olyra latifolia*, and *Oryza sativa*.

**Figure 3 plants-14-01520-f003:**
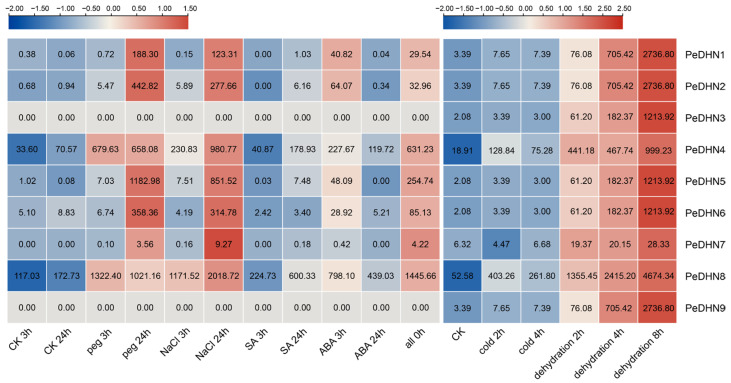
Transcriptome heatmap of PEG, NaCl, SA, ABA, and combined treatments with cold and dehydration. Control (CK): Moso bamboo seedlings were cultured in 1/4 Hoagland nutrient solution (25 ± 2 °C, 70 ± 10% relative humidity, and natural light cycle). Salt stress treatment used nutrient solution containing 200 mM NaCl. Drought stress treatment used nutrient solution containing 25% PEG. SA and ABA treatments used nutrient solutions containing 1 mM SA and 1 μM ABA, respectively. For dehydration treatment, the branches were placed on dry filter paper and treated at room temperature (20 °C and ~50% humidity). For cold treatment, the branches were placed in a dark room at 0 °C.

**Figure 4 plants-14-01520-f004:**
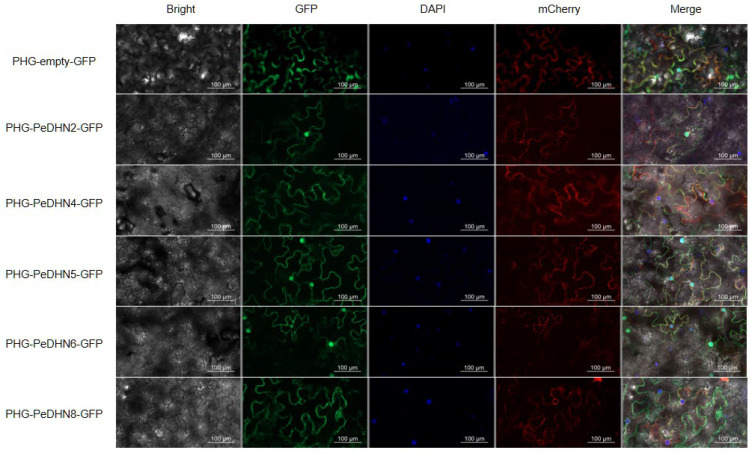
Subcellular localization results. The positive control is the 35S::EGFP empty vector. The 35S::EGFP fusion protein, used as a positive control, is expressed in both the nucleus and plasma membrane. Tobacco leaf cell staining was performed to determine the localization of the fusion protein. Green fluorescence indicates the location of the fusion protein, blue fluorescence marks the nucleus, and red fluorescence marks the plasma membrane.

**Figure 5 plants-14-01520-f005:**
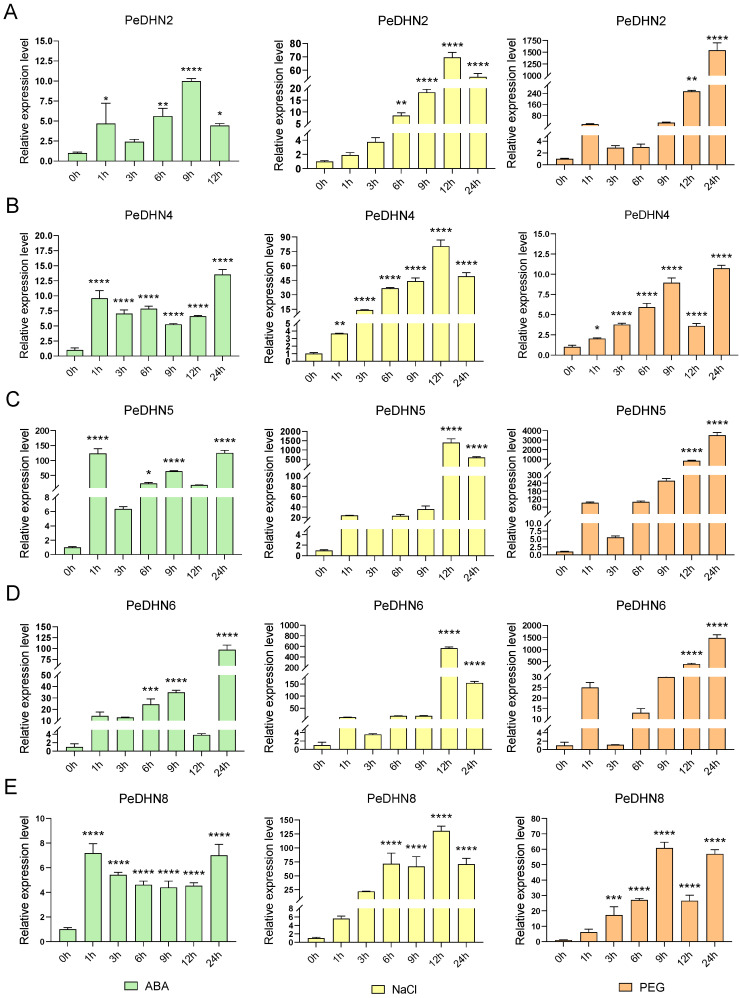
Expression patterns of five *PeDHN* genes in moso bamboo under ABA, NaCl, and PEG stress treatments. Green: ABA treatment, yellow: NaCl treatment, orange: PEG treatment. (**A**) Expression pattern of *PeDHN2* under ABA, NaCl, and PEG treatments. (**B**) Expression pattern of *PeDHN4* under ABA, NaCl, and PEG treatments. (**C**) Expression pattern of *PeDHN5* under ABA, NaCl, and PEG treatments. (**D**) Expression pattern of *PeDHN6* under ABA, NaCl, and PEG treatments. (**E**) Expression pattern of *PeDHN8* under ABA, NaCl, and PEG treatments. *: *p* < 0.05, **: *p* < 0.01, ***: *p* < 0.001, ****: *p* < 0.0001.

**Figure 6 plants-14-01520-f006:**
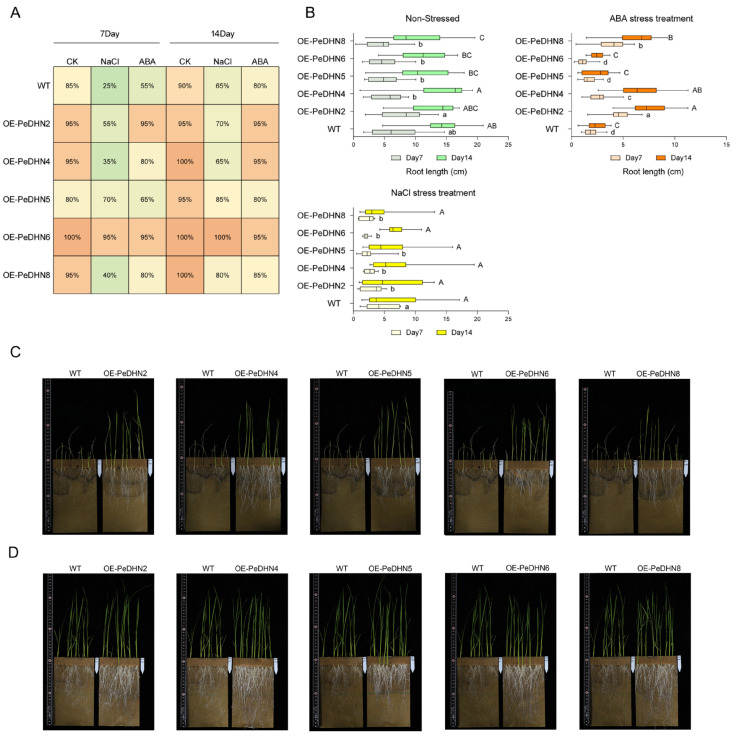
Phenotypic statistics of transgenic rice. (**A**) Heatmap showing the germination rate statistics for the five *OE-PeDHNs* under-untreated, 80 mM NaCl treatment, and 8 μM ABA treatment. Darker shades of orange indicate higher germination rates, while darker shades of green correspond to lower germination rates. (**B**) Root length statistics for the five *OE-PeDHNs* under-untreated, 80 mM NaCl treatment, and 8 μM ABA treatment. Different lowercase letters indicate significant differences between day7 (*p* < 0.05); different uppercase letters indicate significant differences between day14 (*p* < 0.05). (**C**) Phenotypic analysis of transgenic rice after 50 days of continuous NaCl stress treatment. (**D**) Phenotypic analysis of transgenic rice after 50 days of continuous ABA stress treatment.

**Figure 7 plants-14-01520-f007:**
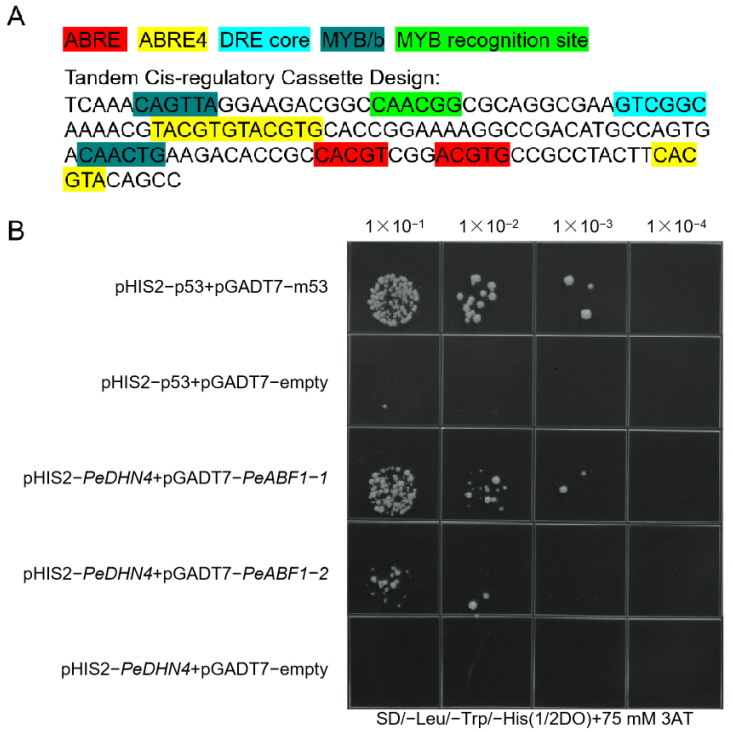
Tandem sequence of promoter cis-acting elements and yeast one-hybrid point-to-point validation. (**A**) Tandem sequence of promoter cis-acting elements of *PeDHN4*. (**B**) Results of yeast one-hybrid point-to-point validation.

**Table 1 plants-14-01520-t001:** Statistical analysis of *PeDHN* YnKnSn types.

Protein	Y Segment	S Segment	K Segment	Subclass
PeDHN1	1 segment VDEYGNP	1 segment SSSSSSS	2 segments EKKGLMDKIKEKLPG, EKKGIKEKIKEKLPG	YK_2_S
PeDHN2	1 segment VDEYGNP	1 segment SSSSSSS	2 segments EKKGLMDKIKEKLPG, EKKGIKEKIKEKLPG	YK_2_S
PeDHN3	1 segment VDEYGNP	1 segment SSSSSSS	2 segments EKKGVMEKIKEKLPG, EKKGIKEKIKEKLPG	YK_2_S
*PeDHN4*	0 segments	1 segment SSSSSSSSS	2 segments EKKGILGKIMEKLPG, EKKGIKEKIKEKLPG	K_2_S
PeDHN5	1 segment VDEYGNP	1 segment SSSSSSS	2 segments EKKGVMEKIKEKLPG, EKKGIKEKIKEKLPG	YK_2_S
PeDHN6	1 segment VDGSGNP	1 segment SSSSSSS	2 segments EKKGIMDKIKEKLPG, EKKGIKEKIKEKLPG	YK_2_S
PeDHN7	2 segments VDQYGNP	1 segment SSSSSSS	2 segments EKKGIMEKIKEKLPG, EKKSIKDKIKEKLPG	Y_2_K_2_S
PeDHN8	0 segment	1 segment SSSSSSSSS	2 segments EKKGILGKIMEKLPG, EKKGIKEKIKEKLPG	K_2_S
PeDHN9	0 segment	1 segment SSSSSSS	2 segments EKKGAMDKIKEKLPG, EKKGIKEKIKEKLPG	K_2_S

## Data Availability

Data are contained within the article and Appendix A.

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
