# Peer review of "Identification and Functional Validation of the PeDHN Gene Family in Moso Bamboo"

_plants, 2025, doi:10.3390/plants14101520_

Round 1

Reviewer 1 Report

Comments and Suggestions for Authors

Suggestions:

The readers should improve the quality of the figure as most of the writing is illegible (specifically for the keys in Figure 1C).

Supplementary Table 1 'proteins' should be added in the table heading. 

Line 104: The author refers to the molecular weights in Da (daltons); however, in the Supplementary Table 1, the SI units are in kDa. Please correct the SI units. 

Line 173: 'Panicum halli' should be italicized. 

Lines 184-185 repeat the follow-up sentence. The author first mentions that 'PeDNH9 showed minimal expression changes', and then in lines 186-187, the author mentions that 'PeDHN9 showed almost no expression under'.

Figure 3: dehydration 4h, the author should remove the 'G'.

Figure 3: The authors mention all the treatment conditions, except cold stress conditions, in the figure legend. Please correct this. 

Supplementary Table 4: The authors mention that PeDHN3 is localized to three locations, however, only two locations are provided in the table. insert nuclease in the table for PeDHN3.

Line 373: 'ZmDHN13' should be italicized. 

Line 512: 'E.coli' should be italicized. 

Line 578: 'Saccharomyces cerevisiae' should be italicized.

Author Response

Point 1. The readers should improve the quality of the figure as most of the writing is illegible (specifically for the keys in Figure 1C).

Response: Thank you for the suggestions. I've optimized the images with enlarged text and higher resolution visuals to improve information clarity.  The specific modifications in the manuscript with revisions_v2 document.

Point 2. Supplementary Table 1 'proteins' should be added in the table heading.

Response: I concur with your recommendation and have revised the title of Supplementary Table 1 to ' Physicochemical Properties of PeDHNs Proteins '.

Point 3. Line 104: The author refers to the molecular weights in Da (daltons); however, in the Supplementary Table 1, the SI units are in kDa. Please correct the SI units.

Response: I sincerely apologize for the oversight. The unit in the table has now been corrected to Da to accurately reflect molecular weight measurements.

Revised text:

Line 112 [ The isoelectric points of these proteins range from 5.23 to 9.85, and their molecular weights vary from 13,223.55 Da to 51,045.33 Da. ]

Point 4. Some nouns that need to be in italics.

Line 173 'Panicum halli' should be italicized.

Line 512 'E.coli' should be italicized.

Line 578 'Saccharomyces cerevisiae' should be italicized.

Response: We sincerely thank the reviewer for their meticulous attention to nomenclatural details. We have italicized the Latin name of the species and have thoroughly checked again other parts that need to be italicized.

Revised text:

Line 185 [ Panicum hallii ]

Line 546 [ E.coli ]

Line 629 [ Saccharomyces cerevisiae ]

Point 5. Lines 184-185 repeat the follow-up sentence. The author first mentions that 'PeDNH9 showed minimal expression changes', and then in lines 186-187, the author mentions that ' PeDHN9 showed almost no expression under '.

Response: My sincere apologies for the oversight, and thank you for your thorough review. I have revised the section accordingly, and your suggestions have significantly enhanced the clarity of the text flow.

Revised text:

Line 203 [ PeDHN9 showed minimal expression changes under PEG, NaCl, SA, ABA, and the combined treatments of PEG, NaCl, SA, and ABA, but expression was significantly increased under dehydration treatment. Additionally, PeDHN3 showed almost no expression under PEG, NaCl, SA, and ABA treatments ]

Point 6.

Figure 3: dehydration 4h, the author should remove the 'G'.

Figure 3: The authors mention all the treatment conditions, except cold stress conditions, in the figure legend. Please correct this.

Response: I sincerely apologize for the oversight and deeply appreciate the reviewer's valuable comments. The figures in the main text have been revised and replaced, with the experimental conditions for dehydration and cold stress treatments now clearly specified in the figure captions. The specific modifications in the manuscript with revisions_v2 document.

Revised text:

Line 207

Figure 3. Transcriptome Heatmap of PEG, NaCl, SA, ABA, and Combined Treatments with Cold and Dehydration. Control (CK): Moso bamboo seedlings were cultured in 1/4 Hoagland nutrient solution (25 ± 2°C, 70 ± 10% relative humidity, and natural light cycle). Salt stress treatment used nutrient solution containing 200 mM NaCl. Drought stress treatment used nutrient solution containing 25% PEG. SA and ABA treatments used nutrient solutions containing 1 mM SA and 1 μM ABA, respectively. For dehydration treatment, place the branches on dry filter paper and treat them at room temperature (20°C and ~50% humidity). For cold treatment, place the branches in a dark room at 0°C.

Point 8. Supplementary Table 4: The authors mention that PeDHN3 is localized to three locations, however, only two locations are provided in the table. insert nuclease in the table for PeDHN3.

Response: I sincerely apologize for the oversight and have conducted a systematic reanalysis of subcellular localization predictions for all PeDHN genes. The corresponding tables and result sections in the manuscript have been revised to reflect these updated computational analyses.

Revised text:

Line 217 [ Subcellular localization predictions from the WOLF PSORT server indicated that, except for PeDHN3, which is localized to the plasma membrane, endoplasmic reticulum, nucleus and golgi apparatus, and PeDHN9, which is localized to the nucleus and chloroplast, the remaining PeDHNs are all localized in the nucleus (Table S4). ]

Table S4.  Subcellular Localization Prediction Results

Protein name

Location

PeDHN1

Nucleus

PeDHN2

Nucleus

PeDHN3

Plasma membrane/Endoplasmic reticulum/Nucleus/Golgi apparatus

PeDHN4

Nucleus

PeDHN5

Nucleus

PeDHN6

Nucleus

PeDHN7

Nucleus

PeDHN8

Nucleus

PeDHN9

Nuclear/Chloroplast

Point 9. Line 373: 'ZmDHN13' should be italicized.

Response: I sincerely apologize for the lack of clarity in the main text. Since ZmDHN13 in this context specifically denotes the protein (not the gene), the non-italicized formatting was initially retained in accordance with nomenclature guidelines. To prevent ambiguity, I will denote it as ZmDHN13 protein in the text.

Revised text: 

Line 398: [ For instance, ZmDHN13 protein in maize is localized to the nucleus, depending on the phosphorylation of the S segment, possibly having DNA repair or protection functions. ]

Reviewer 2 Report

Comments and Suggestions for Authors

Review3605781

Dehydrins (DHNs) play an important role in combating abiotic stresses. Given their significance for stress adaptation and enhancing the productivity, the research of the PeDHN gene family in moso bamboo (Phyllostachys edulis), provides valuable insights for the sustainable cultivation of this multifunctional plant species under adverse climate and environmental conditions. Using integrated bioinformatics approaches, the authors identified nine DHN family members in moso bamboo, analyzed their gene structure, chromosomal locations, phylogeny and distribution of stress-related cis-regulatory elements in their promoter regions. Five genes with higher expression levels under various stress conditions were selected for cloning and analysis of the sub-cellular localization of dehydrins encoded by them. qRT-PCR analysis revealed that the expression of selected genes was responsive to stress treatments with ABA, NaCl, and PEG. The expressing of PeDHNs in transformed japonica rice enhanced rice resistance to NaCl and ABA.  By means of the yeast one-hybrid assay the authors also demonstrated the interaction of PeDHN4 with two ABF1 transcription factors suggesting that signals from abiotic stress may be transmitted to DHN genes through ABF1. The results obtained provide insights into the roles of PeDHNs in enhancing plant tolerance to abiotic stresses and suggest potential mechanisms for their function.

While evaluating the overall positive nature of the study, I would like to draw the authors' attention to a number of noticed shortcomings.

  1. Fig1 Lines 97-98 “distribution of stress-related cis-regulatory elements in the 2000 bp upstream of the PeDHNs promoter”. Obviously, the authors meant “2000 bp upstream of the ATG initiation codon”
  2. “2.4 Subcellular Localization of DHN Gene Family Members” The authors analyzed the localization of proteins encoded by DHN Gene Family Members, since the genes cannot be localized in plasma membrane
  3. Fig 5 Line 251 “Stress Treatments. Light green: control group (untreated)”. There are no controls in the figure.
  4. Lines 326-332 and 449-484. Incorrect text wrapping
  5. 7 Line 337“Promoter Cis-Acting Element Tandem”

Line 338 “Design of the PeDHN4 promoter cis-acting element tandem.:

Probably, the authors meant “promoter  tandem cis-acting elements”

  1. Lines 595-596 “preliminary validation suggests that PeDHN4 may regulate the response to salt and alkali stress through the ABA signaling pathway”. No data on alkali stress or analysis of ABA signaling mutants were presented

Author Response

Dehydrins (DHNs) play an important role in combating abiotic stresses. Given their significance for stress adaptation and enhancing the productivity, the research of the PeDHN gene family in moso bamboo (Phyllostachys edulis), provides valuable insights for the sustainable cultivation of this multifunctional plant species under adverse climate and environmental conditions. Using integrated bioinformatics approaches, the authors identified nine DHN family members in moso bamboo, analyzed their gene structure, chromosomal locations, phylogeny and distribution of stress-related cis-regulatory elements in their promoter regions. Five genes with higher expression levels under various stress conditions were selected for cloning and analysis of the sub-cellular localization of dehydrins encoded by them. qRT-PCR analysis revealed that the expression of selected genes was responsive to stress treatments with ABA, NaCl, and PEG. The expressing of PeDHNs in transformed japonica rice enhanced rice resistance to NaCl and ABA. By means of the yeast one-hybrid assay the authors also demonstrated the interaction of PeDHN4 with two ABF1 transcription factors suggesting that signals from abiotic stress may be transmitted to DHN genes through ABF1. The results obtained provide insights into the roles of PeDHNs in enhancing plant tolerance to abiotic stresses and suggest potential mechanisms for their function.

While evaluating the overall positive nature of the study, I would like to draw the authors' attention to a number of noticed shortcomings.

Point 1. Fig1 Lines 97-98 “distribution of stress-related cis-regulatory elements in the 2000 bp upstream of the PeDHNs promoter”. Obviously, the authors meant “2000 bp upstream of the ATG initiation codon”

Response: Thank you for your insightful comment. We agree with your suggestion and have revised the sentence accordingly.

Revised text: 

Line 105 [ Distribution of stress-responsive cis-regulatory elements involved in abiotic stress tolerance in the 2000 bp upstream of the ATG initiation codon of the PeDHNs promoter. ]

Point 2. “2.4 Subcellular Localization of DHN Gene Family Members”

The authors analyzed the localization of proteins encoded by DHN Gene Family Members, since the genes cannot be localized in plasma membrane

Response: I'm sorry for the confusion, but I've revised the relevant content.

Revised text:

Line 217 [ Subcellular localization predictions from the WOLF PSORT server indicated that, except for PeDHN3, which is localized to the plasma membrane, endoplasmic reticulum, nucleus and golgi apparatus, and PeDHN9, which is localized to the nucleus and chloroplast, the remaining PeDHNs are all localized in the nucleus (Table S4). ]

Point 3. Fig 5 Line 251 “Stress Treatments. Light green: control group (untreated)”. There are no controls in the figure.

Response: I sincerely apologize for the lack of clarity in the content. I have revised the section and removed the statement “ Light green: control group (untreated) ” from the figure legend.

Revised text: 

Line 273 [ Figure 5. Expression Patterns of Five PeDHNs Genes in Moso Bamboo Under ABA, NaCl, and PEG Stress Treatments. Green: ABA treatment, yellow: NaCl treatment, orange: PEG treatment. (A) Expression pattern of PeDHN2 under ABA, NaCl, and PEG treatments. ]

Point 4. Lines 326-332 and 449-484. Incorrect text wrapping

Response: We sincerely appreciate the reviewer's attention to manuscript formatting details. I have adjusted the formatting of these text segments. The revised formatting can be viewed in the Word document with revision marks.

Point 5.

Line 337 “ Promoter Cis-Acting Element Tandem ”

Line 338 “ Design of the PeDHN4 promoter cis-acting element tandem. ”

Probably, the authors meant “ promoter tandem cis-acting elements ”

Response: I sincerely apologize for the lack of clarity in conveying my intended meaning, which may have caused confusion. I have revised the aforementioned content to ensure clearer articulation of the concepts.

Revised text:

Line 262 [ Figure 7. Tandem sequence of promoter cis-acting elements and Yeast One-Hybrid Point-to-Point Validation. (A) Tandem sequence of promoter cis-acting elements of PeDHN4. (B) Results of yeast one-hybrid point-to-point validation. ]

Point 6. Lines 595-596 “preliminary validation suggests that PeDHN4 may regulate the response to salt and alkali stress through the ABA signaling pathway”. No data on alkali stress or analysis of ABA signaling mutants were presented.

Response: We sincerely appreciate the reviewer’s critical assessment and acknowledge the limitations in our initial experimental scope. To clarify and strengthen this conclusion, we have implemented the following revisions:

Arid regions experience scarce rainfall and intense evaporation. As soil moisture evaporates, salts like sodium, calcium, and magnesium sulfates and chlorides accumulate in the surface soil due to lack of leaching by rain, causing soil salinization. Obviously, there is a close relationship between aridity and salinization.

Our experiments show that PeDHN4 interacts with two ABF1 transcription factors. Literature indicates ABF1 is ABA-regulated. In our study, ABA-treated transgenic rice grew better than WT, and RT-qPCR showed PeDHN4 upregulation post-ABA treatment. This suggests ABA has a positive feedback effect on PeDHN4. ABA is crucial in plant drought and salinity tolerance, regulating stomatal movement and stress - response genes, and synergizing with JA and SA.

Regarding the lack of ABA signaling mutant analysis, since we haven't performed transient transformation of PeDHN4 into ABA signaling mutants, this part isn't covered. Including such experiments in our future plans could strengthen the evidence that PeDHN4 regulates responses to drought and saline-alkaline stress via the ABA pathway. And I'll include this perspective in the discussion section.

Revised text:

Line 411 [ Arid regions experience scarce rainfall and intense evaporation. As soil moisture evaporates, salts like sodium, calcium, and magnesium sulfates and chlorides accumulate in the surface soil due to lack of leaching by rain, causing soil salinization. Obviously, there is a close relationship between aridity and salinization. ]

Line 454 [ In the future, if we can perform transient transformation of PeDHN4 in rice ABA signaling mutants, we can further reveal the role of PeDHN4 in the ABA metabolic pathway. ]

Dehydrins (DHNs) play an important role in combating abiotic stresses. Given their significance for stress adaptation and enhancing the productivity, the research of the PeDHN gene family in moso bamboo (Phyllostachys edulis), provides valuable insights for the sustainable cultivation of this multifunctional plant species under adverse climate and environmental conditions. Using integrated bioinformatics approaches, the authors identified nine DHN family members in moso bamboo, analyzed their gene structure, chromosomal locations, phylogeny and distribution of stress-related cis-regulatory elements in their promoter regions. Five genes with higher expression levels under various stress conditions were selected for cloning and analysis of the sub-cellular localization of dehydrins encoded by them. qRT-PCR analysis revealed that the expression of selected genes was responsive to stress treatments with ABA, NaCl, and PEG. The expressing of PeDHNs in transformed japonica rice enhanced rice resistance to NaCl and ABA. By means of the yeast one-hybrid assay the authors also demonstrated the interaction of PeDHN4 with two ABF1 transcription factors suggesting that signals from abiotic stress may be transmitted to DHN genes through ABF1. The results obtained provide insights into the roles of PeDHNs in enhancing plant tolerance to abiotic stresses and suggest potential mechanisms for their function.

While evaluating the overall positive nature of the study, I would like to draw the authors' attention to a number of noticed shortcomings.

Point 1. Fig1 Lines 97-98 “distribution of stress-related cis-regulatory elements in the 2000 bp upstream of the PeDHNs promoter”. Obviously, the authors meant “2000 bp upstream of the ATG initiation codon”

Response: Thank you for your insightful comment. We agree with your suggestion and have revised the sentence accordingly.

Revised text: 

Line 105 [ Distribution of stress-responsive cis-regulatory elements involved in abiotic stress tolerance in the 2000 bp upstream of the ATG initiation codon of the PeDHNs promoter. ]

Point 2. “2.4 Subcellular Localization of DHN Gene Family Members”

The authors analyzed the localization of proteins encoded by DHN Gene Family Members, since the genes cannot be localized in plasma membrane

Response: I'm sorry for the confusion, but I've revised the relevant content.

Revised text:

Line 217 [ Subcellular localization predictions from the WOLF PSORT server indicated that, except for PeDHN3, which is localized to the plasma membrane, endoplasmic reticulum, nucleus and golgi apparatus, and PeDHN9, which is localized to the nucleus and chloroplast, the remaining PeDHNs are all localized in the nucleus (Table S4). ]

Point 3. Fig 5 Line 251 “Stress Treatments. Light green: control group (untreated)”. There are no controls in the figure.

Response: I sincerely apologize for the lack of clarity in the content. I have revised the section and removed the statement “ Light green: control group (untreated) ” from the figure legend.

Revised text: 

Line 273 [ Figure 5. Expression Patterns of Five PeDHNs Genes in Moso Bamboo Under ABA, NaCl, and PEG Stress Treatments. Green: ABA treatment, yellow: NaCl treatment, orange: PEG treatment. (A) Expression pattern of PeDHN2 under ABA, NaCl, and PEG treatments. ]

Point 4. Lines 326-332 and 449-484. Incorrect text wrapping

Response: We sincerely appreciate the reviewer's attention to manuscript formatting details. I have adjusted the formatting of these text segments. The revised formatting can be viewed in the Word document with revision marks.

Point 5.

Line 337 “ Promoter Cis-Acting Element Tandem ”

Line 338 “ Design of the PeDHN4 promoter cis-acting element tandem. ”

Probably, the authors meant “ promoter tandem cis-acting elements ”

Response: I sincerely apologize for the lack of clarity in conveying my intended meaning, which may have caused confusion. I have revised the aforementioned content to ensure clearer articulation of the concepts.

Revised text:

Line 262 [ Figure 7. Tandem sequence of promoter cis-acting elements and Yeast One-Hybrid Point-to-Point Validation. (A) Tandem sequence of promoter cis-acting elements of PeDHN4. (B) Results of yeast one-hybrid point-to-point validation. ]

Point 6. Lines 595-596 “preliminary validation suggests that PeDHN4 may regulate the response to salt and alkali stress through the ABA signaling pathway”. No data on alkali stress or analysis of ABA signaling mutants were presented.

Response: We sincerely appreciate the reviewer’s critical assessment and acknowledge the limitations in our initial experimental scope. To clarify and strengthen this conclusion, we have implemented the following revisions:

Arid regions experience scarce rainfall and intense evaporation. As soil moisture evaporates, salts like sodium, calcium, and magnesium sulfates and chlorides accumulate in the surface soil due to lack of leaching by rain, causing soil salinization. Obviously, there is a close relationship between aridity and salinization.

Our experiments show that PeDHN4 interacts with two ABF1 transcription factors. Literature indicates ABF1 is ABA-regulated. In our study, ABA-treated transgenic rice grew better than WT, and RT-qPCR showed PeDHN4 upregulation post-ABA treatment. This suggests ABA has a positive feedback effect on PeDHN4. ABA is crucial in plant drought and salinity tolerance, regulating stomatal movement and stress - response genes, and synergizing with JA and SA.

Regarding the lack of ABA signaling mutant analysis, since we haven't performed transient transformation of PeDHN4 into ABA signaling mutants, this part isn't covered. Including such experiments in our future plans could strengthen the evidence that PeDHN4 regulates responses to drought and saline-alkaline stress via the ABA pathway. And I'll include this perspective in the discussion section.

Revised text:

Line 411 [ Arid regions experience scarce rainfall and intense evaporation. As soil moisture evaporates, salts like sodium, calcium, and magnesium sulfates and chlorides accumulate in the surface soil due to lack of leaching by rain, causing soil salinization. Obviously, there is a close relationship between aridity and salinization. ]

Line 454 [ In the future, if we can perform transient transformation of PeDHN4 in rice ABA signaling mutants, we can further reveal the role of PeDHN4 in the ABA metabolic pathway. ]

Reviewer 3 Report

Comments and Suggestions for Authors

In this manuscript the author investigated “nine DHN family members (PeDHN1–PeDHN9) from moso bamboo, which were classified into Kâ‚‚S-type, YKâ‚‚S-type, and Yâ‚‚Kâ‚‚S-type dehydrins based on their characteristic motifs. Further, the author employed integrated bioinformatics approaches to analyze their gene structure, phylogeny, biological properties, and expression patterns under various stress conditions. Five genes (PeDHN2/4/5/6/8), which may have significant functional roles in moso bamboo, were selected for cloning. The topic selected by the author is interesting and the experiment performed by the author having the significant results but these results are not clearly described. There are some major corrections which the author needs to improved.

Major recommendation:

  • English should improve by a native person. Especially the Material and Method section. The manuscript requires a thorough proofread by a native person whose first language is English.

Abstract

Abstract is well written but the author need to write it better in the following order

  • Background/Introduction (2-3 lines)
  • Results
  • Conclusion (2-3 lines)

Introduction

  • Introduction section is well written but still need a minor modification:
  1. Page 2: Line 48-50: need to right it more clearly and understandable

Material and Methods

This section need to be improved

  • Page 13, Line 432: Kindly provide the proper details, from where you get the seedlings etc.
  • Page 13, Line 435: Don’t start the sentence with “60 uniformly” etc..
  • Page 13, Line 442: The author need to provide the details of transgenic rice preparation because its important for the readers. Another important question is that why the author selects the Rice transgenic plants because he is studying moso Bamboo?
  • Page 13, Line 444: the author used a word “Positive T1 generation” what is the author meaning? Further the author mention on Page 9, line 262; “52 homozygous T2 transgenic lines were successfully generated”. So, the author used T1 or T2 in experiments?
  • Page 14, Line 450-458: The author need to write the complete name first time and then later use the abbreviation e.g. NCBI, SMART, O. sativa etc.
  • Page 14, Line 467-470: In phylogenetic analysis, the author mention 16 different crops and it’s really confusing for me why author don’t use only Arabidopsis, Rice and moso-bamboo? because he starts with these 3 and it’s better to keep only these 3 in whole study.
  • Page 14, Line 473: In synteny analysis again the new species appear…Why?
  • Page 15, Line 507-510: When the author prepared the cDNA then why he amplified them again? Kindly check your method.
  • Page 15, Line 511: Which cloning kit? Kindly add details.
  • Kindly add the references in material and method section.
  • Page 15, Line 546-547: “four moso bamboo tissues” which four tissues? Kindly write the names.

Result:

  • Page 5, line 157-158: On which basis the author selects these all species?
  • Page 5, line 164: Where is Figure 2.B and 2C?
  • Page 5, line 167: In text you write 15 species but why here its 15?
  • Page 5, line 171: I don’t find Brachypodium in above figure 2.
  • Page 5, line 176: The author write “we performed transcriptome data analysis” but I think the author used transcriptomic data from previous study.
  • Page 6, line 200-202; Why only five? What about the rest of genes? For Subcellular localization why you need to base on transcriptomic data and on promoter study analysis?
  • Page 6, line 207-209; “PeDHN4 and PeDHN8, were exclusively localized to the plasma membrane, with no expression observed in the nucleus” but in Page 6, line 197-199 the author write that “these are located in nucleus”?
  • Page 7, line 219-222: The author checked the expression of 5 genes. I think the author need to check expression of all genes, not only 5 genes.
  • Page 8, line 251: the author mention in Figure 5 legends “Light green: control group (untreated)” but I don’t find it in Figure 5.
  • Page 9, line 262 and 266: Fig. S2A and Fig. S2B-F, is not related to vector construction and even don’t show the 52 homozygous T2 transgenic lines.
  • Page 9, line 265: I don’t find 3 replicates in Figures.
  • Page 9, line 290-292: Kindly show the picture of 50 days untreated plants because the height of 15 days and 50 days don’t show much difference.
  • Page 10, Figure 6: Figure 6D: contains same figure in B (OE-PeDHN4) and C (OE-PeDHN5)? I think the author need to correct the figures because it’s not possible that both the genes have same figures.
  • Page 10-11, line 304-325: Need to correct all the figures number
  • Page 11, line 326: On which basis the author selects these six TF?
  • Author used the same WT figure in figure 6. It will be better to use different WT figures.
Comments on the Quality of English Language

English should improve by a native person. Especially the Material and Method section. The manuscript requires a thorough proofread by a native person whose first language is English.

Author Response

In this manuscript the author investigated “nine DHN family members (PeDHN1–PeDHN9) from moso bamboo, which were classified into Kâ‚‚S-type, YKâ‚‚S-type, and Yâ‚‚Kâ‚‚S-type dehydrins based on their characteristic motifs. Further, the author employed integrated bioinformatics approaches to analyze their gene structure, phylogeny, biological properties, and expression patterns under various stress conditions. Five genes (PeDHN2/4/5/6/8), which may have significant functional roles in moso bamboo, were selected for cloning. The topic selected by the author is interesting and the experiment performed by the author having the significant results but these results are not clearly described. There are some major corrections which the author needs to improved.

Point 1.

Major recommendation:

English should improve by a native person. Especially the Material and Method section. The manuscript requires a thorough proofread by a native person whose first language is English.

Response: We sincerely apologize for the linguistic barriers that have hindered the clarity of the manuscript. To address this, we have engaged a certified English editor specializing in scientific writing to comprehensively revise the Materials and Methods section. Additionally, we have undertaken a systematic review of the entire document to rectify any residual grammatical errors or terminological inconsistencies, ensuring enhanced readability and conceptual precision.

Point 2. Abstract

Abstract is well written but the author need to write it better in the following order

Background/Introduction (2-3 lines)

Results

Conclusion (2-3 lines)

Response: Thanks a lot for the reviewers' suggestions. I agree with you and have modified the abstract part of the paper in accordance with your advice.

Revised text:

Line 14 [ As climate change intensifies soil drought and salinization, enhancing the drought and salt tolerance of moso bamboo (Phyllostachys edulis) is urgent. DHN genes are crucial for plant stress responses and have gained attention in plant resistance to drought and salinity. This study identified nine DHN family members (PeDHN1PeDHN9) from moso bamboo, which were classified into Kâ‚‚S-type, YKâ‚‚S-type, and Yâ‚‚Kâ‚‚S-type dehydrins based on their characteristic motifs. We employed integrated bioinformatics approaches to analyze their gene structure, phylogeny, biological properties, and expression patterns under various stress conditions. Five genes (PeDHN2/4/5/6/8), which may have significant functional roles in moso bamboo, were selected for cloning. Subcellular localization experiments showed that YK2S-type dehydrins (PeDHN2/5/6) localized to both the nucleus and the plasma membrane, while K2S-type dehydrins (PeDHN4/8) were exclusively localized to the plasma membrane, indicating functional differentiation. qRT-PCR analysis revealed that the expression of PeDHN2/4/5/6/8 was significantly responsive to stress treatments with ABA, NaCl, and PEG. Additionally, overexpressing these genes in rice significantly enhanced seed germination rates and root development under salt and ABA stress, further confirming that PeDHN2/4/5/6/8 contribute to enhancing plant stress tolerance. Yeast one-hybrid assays demonstrated that two PeABF1 proteins could bind to the promoter of PeDHN4, suggesting that PeDHN4 may regulate stress responses through the ABA signaling pathway. Thus, we can preliminarily infer that the DHN gene family has a close relationship with the ABA metabolic pathway. This research offers insights for moso bamboo cultivation and theoretical foundations for bamboo genetic improvement in stress environments. ]

Point 3. Introduction

Introduction section is well written but still need a minor modification:

Page 2: Line 48-50: need to right it more clearly and understandable

Response: Thank you for your useful suggestions. I agree to revise this part.

Revised text:

Line 52 [ Over 20 years ago, Close et al. elucidated the conserved structural architecture of dehydrins by identifying three signature motifs enriched in tyrosine (Y), serine (S), and lysine (K) residues. This discovery established the YSK nomenclature system. Based on the types and numbers of signature motifs containing these characteristic amino acids, DHNs can be divided into five types: Kn, SKn, KnS, YnKn, and YnSKn. ]

Point Material and Methods

This section need to be improved

Page 13, Line 432: Kindly provide the proper details, from where you get the seedlings etc.

Response: We sincerely appreciate the reviewer's emphasis on material traceability. We've expanded the plant materials section and added more details.

Revised text:

Line 464 [ In this study, we used moso bamboo seeds (collected from Guilin, Guangxi, China) as the research materials and grew them into seedlings in a controlled environment chamber. ]

Page 13, Line 435: Don’t start the sentence with “60 uniformly” etc..

Response: Thank you very much for pointing out my inappropriate wording. I have made the necessary changes and modified the descriptions.

Revised text:

Line 468 [ Sixty seedlings of similar size and growth were selected...... ]

Page 13, Line 442: The author need to provide the details of transgenic rice preparation because its important for the readers. Another important question is that why the author selects the Rice transgenic plants because he is studying moso Bamboo?

Response: Thank you for your suggestion. I have detailed the transgenic rice preparation process. We chose transgenic rice because only Dendrocalamus latiflorus has a stable and efficient transformation system, and it's exclusive to certain labs. Thus, we can't conduct DHN gene function research via transgenic overexpression in Moso bamboo. Among current heterologous transformation systems, rice, being in the same family as bamboo, is the most related. So, we used rice for overexpression, a method widely accepted in Moso bamboo breeding.

Revised text:

Line 593 [ The constructed plant expression vector plasmids were heat-shock transformed into Agrobacterium EHA105 competent cells (Boyuan, Wuhan, China) and sent to Boyuan BioTech Co. (Wuhan, China) for genetic transformation of japonica rice (cv. Nipponbare). Choose mature rice grains that are free from mold contamination and have intact embryos. Disinfect the grains in 75% alcohol for 1 minute, then rinse with sterile water. Next, disinfect them in 15% sodium hypochlorite for 20 minutes, followed by three rinses with sterile water, each lasting 1 minute. Inoculate the sterilized grains onto callus induction medium and incubate at 26℃ under a 16-hour photoperiod for 20 days. Infect the developed calli with Agrobacterium competent cells suspension carrying the target plasmid, then transfer them to screening medium and incubate at 26℃ in the dark for 20 to 30 days. Move the positive callus to secondary screening medium and culture under the same conditions for an additional 7 to 10 days. Transfer the callus to differentiation medium and culture at 25 to 27℃ with light for 15 to 20 days. Once shoots reach 2 to 5 centimeters in length, transfer them to rooting medium and culture at 30℃ with light for 7 to 10 days. Finally, gel electrophoresis analysis with hygromycin-specific detection primers confirmed 15 transgenic japonica rice seedlings, all exhibiting positive PCR amplification bands. Seeds collected from these plants after full cultivation cycles were subsequently used for functional validation studies. ]

Page 13, Line 444: the author used a word “Positive T1 generation” what is the author meaning? Further the author mention on Page 9, line 262; “52 homozygous T2 transgenic lines were successfully generated”. So, the author used T1 or T2 in experiments?

Response: I sincerely apologize for the mistake in the statement at Page 9, Line 262. I have carefully checked it and corrected it to T1, which indicates the first-generation progeny obtained by planting the initial transgenic plants.

Revised text:

Line 283: [ After hygromycin selection and PCR validation with marker primers, 52 homozygous T1 transgenic lines were successfully generated (Fig. S1A). ]

Page 14, Line 450-458: The author need to write the complete name first time and then later use the abbreviation e.g. NCBI, SMART, O. sativa etc.

Response: Thank you for your valuable suggestion. We have revised the text at Page 14, Lines 450-458. The complete names of terms such as NCBI, SMART, and O. sativa have been written out in full upon their first mention, with the abbreviations provided in parentheses. Subsequent references use only the abbreviations.

Page 14, Line 467-470: In phylogenetic analysis, the author mention 16 different crops and it’s really confusing for me why author don’t use only Arabidopsis, Rice and moso-bamboo? because he starts with these 3 and it’s better to keep only these 3 in whole study.

Response: When constructing the phylogenetic tree, I didn't limit myself to the protein sequences of Arabidopsis thaliana, Oryza sativa, and Phyllostachys edulis. This is because Arabidopsis has only four identified DHN genes, and Oryza sativa has five, which is not enough to clearly show the differences among DHN genes. So, I included 13 DHN genes identified in Zea mays and 11 highly similar genes from eight species obtained by aligning PeDHNs protein with the NCBI library. This increased the number of protein sequences to enhance the resolution of the phylogenetic analysis and improve the results' reliability.

Page 14, Line 473: In synteny analysis again the new species appear…Why?

Response: After reviewing extensive literature, I found that in Moso bamboo research, analyses often involve bamboo species and Oryza sativa, a closely related grass. In my synteny analysis, I initially tried model species like Arabidopsis thaliana, Oryza sativa, Zea mays, Populus and Physcomitrium patens. However, due to genome file differences, some species showed no synteny with Moso bamboo, or their varying chromosome lengths made it hard to display synteny on the same figure. After multiple attempts, I chose Oryza sativa, Bonia amplexicaulis and Olyra latifolia for analysis to better show synteny between Moso bamboo, rice, and other bamboo species.

Page 15, Line 507-510: When the author prepared the cDNA then why he amplified them again? Kindly check your method.

Response:I apologize for the confusion. I've made the necessary revisions to this part. I used real-Time quantitative reverse transcription PCR (RT-qPCR). Using the abbreviation RT-qPCR is more standard and can reduce confusion. I have carefully checked and modified the content related to RT-qPCR accordingly.

Revised text:

Line 573 [ 4.7. Real-Time Quantitative Reverse Transcription PCR of Five PeDHN Genes Following Stress Treatment

Seven-month-old moso bamboo seedlings were treated with Kimura's nutrient solution alone, and nutrient solutions containing 80 mM ABA, 100 mM NaCl, or 20% PEG for 0, 3, 6, 9, 12, and 24 h. At each time point, the second leaf from the top was collected for real-time quantitative reverse transcription PCR(RT-qPCR) analysis. Quantitative PCR primers for the PeDHN genes were designed using Primer Premier v.6 (Table S6) [40]. RT-qPCR was performed using TB Green & Premix Ex TaqTM II (TaKaRa) on a qTOWER RT-PCR system (Analytik, Jena, Germany) ]

Page 15, Line 511: Which cloning kit? Kindly add details.

Kindly add the references in material and method section.

Response: I'm really sorry for not clearly indicating the cloning kit in the article. I've already made the revisions in the materials and methods section.

Revised text:

Line 563 [ The target sequence was inserted into the pCE2TA/Blunt-Zero vector (Vazyme, Nanjing, China) using 5 min TA/Blunt-Zero Cloning Kit (Vazyme, Nanjing, China) in a metal bath at 25℃. ]

Page 15, Line 546-547: “four moso bamboo tissues” which four tissues? Kindly write the names.

Response: We sincerely appreciate the reviewer's meticulous reading. We have revised this statement in the Materials and Methods section to precisely reflect the experimental design.

Revised text:

Line 581 [ GAPDH was used as the reference gene, and the relative gene expression in moso bamboo leaves under three treatments and six time points was analyzed using the 2-ΔΔCT method. ]

Result:

Page 5, line 157-158: On which basis the author selects these all species?

Response: We sincerely thank the reviewer for their questions. Our species selection was primarily based on the following reasons:

(1) Phylogenetic Tree Analysis

Arabidopsis is the most representative dicot model plant, while rice and maize are highly representative monocot model plants within the Poaceae family. Other Poaceae species were selected based on high similarity rankings obtained by comparing moso bamboo DHN protein sequences with those in the NCBI public database. Using only model species could not adequately distinguish the differences within the moso bamboo DHN gene family, hence supplementary species were added. Other bamboo species were excluded because there is currently no literature supporting the identification or study of DHN genes in other bamboos.

(2) Synteny Analysis Across Species

Initial analysis included Arabidopsis thaliana, Oryza sativa, Zea mays, Populus, Physcomitrium patens, Bonia amplexicaulis and Olyra latifolia. However, some species showed no synteny with Moso bamboo and were excluded. Others had chromosome lengths that differed too much from Moso bamboo to be displayed effectively in a single figure.

Finally, Based on visualization effectiveness, representativeness, and existing literature on moso bamboo synteny analysis (which predominantly uses rice and other bamboo genomes), we ultimately selected rice, Olyra latifolia, and Bambusa ventricosa for synteny analysis.

Page 5, line 164: Where is Figure 2.B and 2C?

Response: We sincerely apologize for the omission of a section due to formatting errors during manuscript preparation; the missing content has now been fully restored in the revised manuscript.

Revised text:

Line 175 [ Collinearity within the moso bamboo genome as well as between other bamboo genomes and the rice genome was analyzed using the MCScanX software. The results showed that there are two pairs of collinear genes (PeDHN8 and PeDHN4, PeDHN5 and PeDHN6) of PeDHNs in moso bamboo (Fig. 3B), which may be related to the two rounds of genome duplication events in moso bamboo [16]. Interspecific collinearity analysis was also performed on Bonia amplexicaulis, Olyra latifolia and O.sativa. There are three pairs of collinear genes between moso bamboo and O. latifolia, five pairs between moso bamboo and B. amplexicaulis, and 11 pairs between moso bamboo and rice (Fig. 2C). ]

Page 5, line 167: In text you write 15 species but why here its 15?

Page 5, line 171: I don’t find Brachypodium in above figure 2.

Response: I'm truly sorry for this significant error and the questions it raised. While trying different species and gene combinations to build the phylogenetic tree, I got confused when writing the paper, which resulted in an incorrect number of species being reported. I have now checked the protein sequence files used for the phylogenetic tree construction, confirmed all species and genes involved, and corrected all related errors in the text.

Revised text:

Line 162 [ To obtain the genomic characteristics and evolutionary information of the moso bamboo DHN gene family, we constructed a phylogenetic tree using homologous genes from the dicotyledonous model plant Arabidopsis thaliana, the monocotyledonous model plants Oryza sativa and Zea mays , and eight additional Poaceae species. ]

Line 182 [ Figure 2. Phylogenetic and Synteny Analysis of Moso Bamboo and Related Species. (A) Phylogenetic analysis of 11 species based on moso bamboo using the Neighbor-Joining method. (B) Synteny analysis within moso bamboo species. (C) Synteny analysis between moso bamboo and Bonia amplexicaulis, Olyra latifolia, and Oryza sativa.

Abbreviations for all species:

Hv - Hordeum vulgare, Ob - Oryza brachyantha, Og - Oryza glaberrima, Pa - Phragmites australis, Si - Setaria italica, Ta - Triticum aestivum, Td - Triticum dicoccoides, Tu - Triticum urartu. ]

Page 5, line 176: The author write “we performed transcriptome data analysis” but I think the author used transcriptomic data from previous study.

Response: We sincerely appreciate the reviewer’s meticulous attention to methodological transparency. Here, I need to make some clarifications: The transcriptome data used is indeed from other researchers' previously published work. However, they have not conducted any targeted or further analysis on the DHN gene family in Moso bamboo. Moreover, some data was uploaded to NCBI public database without associated published papers. 

Page 6, line 200-202; Why only five? What about the rest of genes? For Subcellular localization why you need to base on transcriptomic data and on promoter study analysis?

Response: We appreciate the reviewer's inquiry regarding our gene selection strategy. The prioritization of PeDHN2/4/5/6/8 for functional validation was based on a three-tiered rationale: Firstly, by analyzing promoter cis-acting elements, it was found that DHN3 and DHN9 have fewer stress-related cis-acting elements than other DHN genes. Secondly, under various stress treatments, some genes like DHN3, DHN7, and DHN9 showed no significant changes in expression in transcriptomic data. Thirdly, due to the high difficulty in gene cloning, DHN1 couldn't be cloned. So, finally, only PeDHN2/4/5/6/8 could be studied.

Page 6, line 207-209; “PeDHN4 and PeDHN8, were exclusively localized to the plasma membrane, with no expression observed in the nucleus” but in Page 6, line 197-199 the author write that “these are located in nucleus”?

Response: We appreciate the reviewer’s careful reading. We would like to clarify the apparent discrepancy: while the WOLF PSORT server predicted that PeDHN4 and PeDHN8 were localized to the cell nucleus, subsequent experimental validation via confocal microscopy demonstrated conclusively that these proteins are not nuclear-localized. Although in WOLF PSORT server predictions provide preliminary insights into subcellular targeting, they may not fully account for post-translational modifications or dynamic cellular contexts, as observed here. These findings underscore the necessity of empirical validation in determining protein localization.

Page 7, line 219-222: The author checked the expression of 5 genes. I think the author need to check expression of all genes, not only 5 genes.

Response: I strongly agree with your point of view. During the actual experiment, I also tried to perform RT-qPCR on the remaining genes. However, the expression stability of several other genes in the fluorescence detection wasn't satisfactory. I think this is partly related to primer design. Also, as the transcriptomic data shows, the low expression levels of these genes may make it hard to detect fluorescence, thus causing highly unstable data. So, for now, I've only carried out RT-qPCR on the five DHN genes.

Page 8, line 251: the author mention in Figure 5 legends “Light green: control group (untreated)” but I don’t find it in Figure 5.

Response: I'm sorry for the error caused by my carelessness. I've revised this part.

Revised text:

Line 273 [ Figure 5. Expression Patterns of Five PeDHNs Genes in Moso Bamboo Under ABA, NaCl, and PEG Stress Treatments. Green: ABA treatment, yellow: NaCl treatment, orange: PEG treatment. ]

Page 9, line 262 and 266: Fig. S2A and Fig. S2B-F, is not related to vector construction and even don’t show the 52 homozygous T2 transgenic lines.

Page 9, line 265: I don’t find 3 replicates in Figures.

Response: Sorry for the poor reading experience. There was an error in the figure number, which should be Fig. S1. The phrase "3 replicates" was also a typing error. I've made the corrections.

Revised text:

Line 285 [ To assess seed germination and early seedling growth of the transgenic lines under various stress conditions, we cultured the transgenic rice in dual-germination bags using hydroponics. Ten rice seeds were placed in each germination bag, and NaCl (80 mM) and ABA (8 µM) were used as stress treatments. The experimental groups were labeled as OE-PeDHN2/4/5/6/8 and WT (Wild-Type).(Fig. S1 B-F). ]

Page 9, line 290-292: Kindly show the picture of 50 days untreated plants because the height of 15 days and 50 days don’t show much difference.

Response: Thank you for your careful observation. I need to clarify regarding your question. Fig. 6 C and D show the conditions after 50 days of treatment. As for the small height difference of rice seedlings after 50 days, this may be due to prolonged stress, causing the aerial parts to wither from the leaf tips and some inevitable breakage. However, by observing the root system underground, it's evident that the conditions differ from those at 15 days.

Page 10, Figure 6: Figure 6D: contains same figure in B (OE-PeDHN4) and C (OE-PeDHN5)? I think the author need to correct the figures because it’s not possible that both the genes have same figures.

Response: Sorry, I don't understand your point. I've carefully compared the photos in Figure 6, and each one with WT as a control is different.

Author used the same WT figure in figure 6. It will be better to use different WT figures.

Response: I agree that using different WT controls enhances experimental perfection. But cultivating five transgenic rice lines with WT and experimental repeats is labor - intensive. To reduce workload while ensuring effectiveness, I didn't set up WT controls for each transgenic line. However, all transgenic rice stress experiments were conducted simultaneously. I used two germination bags with 10 seeds each (20 seeds in total), ensuring some repeatability and maintaining experimental reliability.

Round 2

Reviewer 3 Report

Comments and Suggestions for Authors

Kindly find the word document.

Author Response

Point 1. Native English speakers are missing in the acknowledgement section, and there are still grammatical errors.

Response: We have had the draft professionally edited by a native English speaking team. Then we carefully reviewed the entire text for both grammar and technical vocabulary and added the English editing service to our Acknowledgments.

Revised text:

Line 665 [ Finally, I would like to thank ProNet Biotech Co., Ltd. for their technical support and MJEditor for linguistic assistance during the preparation of this manuscript. ]

Point 2. The author replied that “We chose transgenic rice because only Dendrocalamus latiflorus has a stable and efficient transformation system, and it's exclusive to certain labs. Thus, we can't conduct DHN gene function research via transgenic overexpression in moso bamboo. Among current heterologous transformation systems, rice, being in the same family as bamboo, is the most related. If the author has a reference for this, it should be included in the article, as it is essential for the readers' understanding.

Response: We sincerely appreciate your valuable suggestions. In response, we have incorporated the recommended content into the Introduction section and supplemented it with multiple studies demonstrating the established use of Oryza sativa as a transgenic model system in moso bamboo research.

Revised text:

Line 37 [ Moso bamboo (Phyllostachys edulis), a representative bamboo species, is one of the most multifunctional and renewable resources in the plant kingdom, often referred to as the "green gold of the forest". In recent years, mounting environmental pressures have driven extensive research into stress-resistant genes and tolerance mechanisms in moso bamboo. Although multiple candidate genes have been identified, the absence of an effective genetic transformation system has hindered functional validation, impeding research and breeding advancements [2,3]. This limitation has led researchers to employ rice—a closely related species with an established transformation system—as a surrogate model for functional validation studies [4-7]. ]

Point 3. The author replied “When constructing the phylogenetic tree, I didn't limit myself to the protein sequences of Arabidopsis thaliana, Oryza sativa, and Phyllostachys edulis. This is because Arabidopsis has only four identified DHN genes, and Oryza sativa has five, which is not enough to clearly show the differences among DHN genes. So, I included 13 DHN genes identified in Zea mays and 11 highly similar genes from eight species obtained by aligning PeDHNs protein with the NCBI library. I'm sorry, but I'm not satisfied with this explanation. If approached this way, the author may find many similar sequences across multiple crops, which would imply adding numerous sequences to the article. Therefore, I suggest that the author stick to the crops already mentioned in the article, except for Arabidopsis, as it is a model plant.

Response: I've accepted your suggestions, reconducted the phylogenetic tree analysis and remade relevant figures. Meanwhile, I've also modified the text about the analysis results.

Revised text:

Line 167 [ To obtain the genomic characteristics and evolutionary information of the moso bamboo DHN gene family, we constructed a phylogenetic tree using homologous and genome-annotated protein sequences from the dicot model plant Arabidopsis thaliana and the monocot model plant Oryza sativa. The goal was to explore the relationship between PeDHNs in moso bamboo and those in other species, thereby predicting their potential functions. Phylogenetic analysis revealed that PeDHNs can be divided into four specific subgroups. PeDHN9 formed an independent cluster. PeDHN7 clustered with AtLEA, while PeDHN5 exhibited close phylogenetic affinity with OsLEA25. PeDHN4 and PeDHN8 were grouped together and may share functional similarity with OsLIP9. Additionally, PeDHN1, PeDHN2, PeDHN3, PeDHN5, and PeDHN6 constituted another cluster. Notably, PeDHN3 appears to be functionally analogous to OsLEA25, whereas PeDHN6 may exhibit functional parallels with OsRab16A. Collectively, most DHN genes in moso bamboo demonstrated closer evolutionary relationships with Oryza sativa within the Poaceae family. (Fig. 2A). ]

Line 503 [ ClustalX was used to align the PeDHN sequences with DHN sequences from Arabidopsis thaliana and Oryza sativa. ]

Point 4. The author replied “After reviewing extensive literature, I found that in moso bamboo research, analyses often involve bamboo species and Oryza sativa, a closely related grass. In my synteny analysis, I initially tried model species like Arabidopsis thaliana, Oryza sativa, Zea mays, Populus and Physcomitrium patens. However, due to genome file differences, some species showed no synteny with moso bamboo, or their varying chromosome lengths made it hard to display synteny on the same figure. After multiple attempts, I chose Oryza sativa, Bonia amplexicaulis and Olyra latifolia for analysis to better show synteny between moso bamboo, rice, and other bamboo species”. My response is the same as above: the author should stick to the crops mentioned in the article, except for Arabidopsis, as it is a model plant. Alternatively, it may be better not to include it in the article or to use alternate method etc

Response: Please accept my sincere apologies. While attempting to implement your suggestions, we re-examined the collinearity analysis between moso bamboo and Arabidopsis thaliana. However, our updated analysis revealed no collinear gene pairs between moso bamboo DHN genes and the dicot model plant A. thaliana (as shown in the following figure). Therefore, we have decided to retain the original collinearity analysis results in our study, which contained moso bamboo, two species of the Bamboo subfamily and the model plant of the Gramineae family, rice.

To clarify the rationale for species selection in our collinearity analysis, we have referenced published collinearity analyses of moso bamboo across related species from prior studies. This evidence substantiates the validity of our experimental design.

In the study titled 'A moso bamboo transcription factor, Phehdz1, positively regulates the drought stress response of transgenic rice', collinearity analysis was performed using Oryza sativa, Poaceae species, and other bamboo species. The figure below displays the microsynteny analysis of Phehdz genes among moso bamboo, Brachypodium distachyon, Oryza sativa, Sorghum bicolor, and Zea mays.

Furthermore, a Master's thesis entitled 'Identification of the FBA Family in Phyllostachys edulis and Functional Characterization of PeFBA8 and PeFBA6 in Moso Bamboo' similarly incorporated Oryza sativa and other bamboo species in its genomic comparative analysis. The figure below presents the collinearity analysis of PeFBAs with Bonia amplexicaulis and Oryza sativa. Collinear relationships between PeFBAs and BamFBAs are depicted with blue lines, whereas those between PeFBAs and OsFBAs are represented by red lines.

Therefore, we ultimately selected moso bamboo, Bonia amplexicaulis, Olyra latifolia, and Oryza sativa for collinearity analysis.

Reference:

Gao Y, Liu H, Zhang K, et al. A moso bamboo transcription factor, Phehdz1, positively regulates the drought stress response of transgenic rice[J]. Plant cell reports, 2021, 40: 187-204.

Li Tiankuo, Identification of the FBA Family in Phyllostachys edulis and Functional Characterization of PeFBA8 and PeFBA6 in Moso Bamboo[D].Chinese Academy of Forestry, 2024, 20.

Point 5. Page 10, Figure 6D: contains same figure in B (OE-PeDHN4) and C (OE-PeDHN5)? I think the author need to correct the figures because it’s not possible that both the genes have same figures. Author Response: Sorry, I don't understand your point. I've carefully compared the photos in Figure 6, and each one with WT as a control is different. If I am not mistaken, these two figures appear to be the same. I kindly request the author to recheck and correct this mistake.

Response: We sincerely apologize for the confusion caused by the figure assembly error and deeply appreciate the reviewer’s meticulous attention to detail. Upon re - examining the original image files, we have now re - edited and changed the figures.

Other mistakes

  • Page 1; Line 30-32: “we can preliminarily infer that the DHN gene family has a close relationship with the ABA metabolic pathway”. The author need to correct the sentence because I don’t find relation with ABA metabolic pathway.

Line 30 [ Thus, these findings demonstrate that PeDHN2/4/5/6/8 are closely related to the response  to drought and saline-alkali environmentsof moso bamboo. ]

  • Page 2, Line 51: Close et al., ? kindly correct the reference

Line 53 [ Over 20 years ago, Close elucidated the conserved structural architecture of dehydrins by identifying three signature motifs enriched in tyrosine (Y), serine (S), and lysine (K) residues. This discovery established the YSK nomenclature system. ]

  • Page 17, Line 593-611. Write the method in article format, currently it looks like a thesis format.

Line 596 [ The plant expression vectors were introduced into Agrobacterium tumefaciens EHA105 (Boyuan, Wuhan, China) via heat-shock transformation for rice genetic transformation. Mature japonica rice (Oryza sativa cv. Nipponbare) seeds were sequentially surface-sterilized in 75% ethanol (1 min) and 15% sodium hypochlorite (20 min), followed by three sterile water rinses. Sterilized seeds were cultured on callus induction medium (26°C, 16 h photoperiod, 20 days). Resulting calli were infected with Agrobacterium suspensions containing target plasmids and subsequently transferred to screening medium (26°C, dark, 20 - 30 days). Positive calli underwent secondary screening (7 - 10 days) prior to differentiation on specific medium (25-27°C, light, 15-20 days). Shoots reaching 2-5 cm in height were transferred to the rooting medium (30°C, light, 7-10 days). PCR amplification using hygromycin resistance gene-specific primers with electrophoretic verification identified 15 transgenic lines exhibiting clear bands. ]

  • Page 17, Line 581. “GAPDH was used as the reference gene”, Kindly write the gene number

Line 584 [ GAPDH (PH02Gene12335) was used as the reference gene, and the relative gene expression in moso bamboo leaves under three treatments and six time points was analyzed using the 2-ΔΔCT method. ]

  • Page 5, Line 178: I think its “2B” figure not “3B”. Kindly correct

Line 184 [ The results showed that there are two pairs of collinear genes (PeDHN8 and PeDHN4, PeDHN5 and PeDHN6) of PeDHNs in moso bamboo (Fig. 2B), which may be related to the two rounds of genome duplication events in moso bamboo [22]. ]

Finally, we would like to express our sincere gratitude once again to the reviewers for their constructive suggestions, which have greatly improved the quality of this manuscript. A Word document with tracked changes has been attached separately for your review.
